# Integrated Seismic Risk Assessment in Nepal

Sanish Bhochhibhoya[1], Roisha Maharjan[1]

[1]Department of Civil Engineering, Pulchowk Campus, Lalitpur, Nepal

*Correspondence to*: Roisha Maharjan (072bce120.roisha@pcampus.edu.np)

**Abstract.** Seismic risk analysis is necessary to mitigate the potential losses resulting from future earthquakes and supplement scientific risk management. In order to assist systematic evaluation and management of risk, it is indispensable to interpret risk in terms of social and economic consequences due to hazardous events like earthquakes. There is an interrelationship between hazards, physical risk, and the social characteristics of populations. Therefore, based on the existing studies focusing on each of these aspects, this paper presents the integrated seismic risk assessment along the sub divisional administrative units of Nepal using 2011 census data. The administrative unit, provinces, are subdivided into districts, and each district into municipalities and Village Development Committee (VDC). The districts, municipalities, and VDCs were considered as our study units. In this paper, the physical or seismic risk was evaluated from the exposure model, hazard curves, and the vulnerability model of the country whereas, the social vulnerability was assessed using Social Vulnerability Index (SoVI) methods. To formulate the physical risk, the assets used were five types of buildings under the exposure model. This model was combined with the physical vulnerability functions of the building and the hazard curves of the country. The result of the physical risk has been presented as Annual Average Loss (AAL). Similarly, among 92 social vulnerability variables, 54 variables were reduced to seven weighted parameters using Principal Component Analysis (PCA). The scores of a total of 45 parameters were used to evaluate the SoVI index, which was further combined with the physical risk to evaluate integrated risk. The results showed that populated cities like Kathmandu, Hetauda, and Janakpur have a highly integrated risk index. Similarly, the Terai region bordering neighbour India and some parts of the central Hilly region are highly vulnerable, while most parts of the Mountainous region in the central and eastern regions are least vulnerable. The results from the present study can be utilized as a part of a comprehensive risk management framework at the district level to recuperate and recover from earthquakes.

Keywords: Socio-economic vulnerability, Physical vulnerability, Seismic risk, Hazard mitigation, OpenQuake, Principal Component Analysis (PCA), Integrated risk

## 1 Introduction

Nepal is one of the seismically active regions in the world with a long record of destructive earthquakes. This is due to the intrinsic geological features with high exposure to earthquakes causing potential severe consequences. Most devastating earthquakes were reported in 1255, 1408, 1681, 1803, 1810, 1833, 1934, and 1988 (Pandey et al., 1999). Table 1 shows the number of deaths caused by the earthquake and poor hazard management in Nepal. From 2000 to 2015, 192 earthquakes greater than moment magnitude ($M_w$) 5, 14 earthquakes greater than $M_w$ 6, and one earthquake greater than 7.5 took place in Nepal. Among these earthquakes, the most recent one in 2015 killed 8948 people, destroyed world heritage sites, and caused estimated damage of 10 billion dollars with a moment magnitude of $M_w$ 7.8 (Mori et al., 2020). Around the globe, the impact of the seismic hazard has escalated due to increased population density, unmanaged urbanization, and other socio-economic parameters (Pachauri et al., 2014). The destruction or disaster is the combination of exposure to natural hazards and conditions of vulnerability characterized by the place and the inability to mitigate the negative repercussions (UNISDR, 2009; Rao et al., 2020). Although natural hazards are not escapable,

hazard mitigation, vulnerability assessment, and their integration can significantly reduce the negative effect and aid in recovery (Frigerio et al., 2016).

Vulnerability is the key element and prerequisite for mitigating disaster and facilitating hazard resilient communities (Guo and Kapucu, 2020; Douglas, 2007). The core elements of vulnerability include resilience, exposure, and sensitivity(Cutter et al., 2003). The biophysical and natural components and the built-in environment under vulnerability have been meticulously examined; however, the social aspects of the vulnerability are highly undermined (Mileti, 1999). As a result, the loss estimation reports are usually unable to reflect social losses. It is imperative to include social

vulnerability while assessing the natural hazards and their losses. According to Cutter et al. (2003), social vulnerability can be evaluated using the social vulnerability index (SoVI). For each country, SoVI is the corresponding measure of overall social vulnerability. Assessment of vulnerability and its mitigation necessitates the understanding of various factors like social, economic, and political contexts (Hewitt, 2007). SoVI analysis uses an all-inclusive framework where each factor is viewed to play an equal contribution to the country's vulnerability (Cutter et al., 2003). This concept has

been applied in geographical and social contexts around the world in the US (South Carolina) (Schmidtlein et al., 2011), Iran (Alizadeh et al., 2018) and Bangladesh (Rahman et al., 2015). Studying social vulnerability identifies the sensitive areas and populations that are prone to high risk and are less likely to acclimatize and recover from natural disasters.

To diminish the losses from natural and man-made hazards, individuals and policymakers need to be responsible and create a resilient community to combat the consequences caused by disasters. Policymakers should focus on mitigating

future risks and make endeavours toward sustainable development. The knowledge transfer between individuals and policymakers is very constrained as hazard and risk assessment put limited focus on social components (Borden et al., 2007). In Nepal, the quantitative assessment of social vulnerability associated with seismic hazards are less owing to the lack of social data for analysis and mapping (Aksha et al., 2019). Many studies in past focused on geographical/physical vulnerability assessment of hazards like a flood (Dixit, 2003), landslides (Malakar, 2014), and extreme weather events

(Shrestha, 2005). The study by Mainali and Pricope (2019) and Aksha et al. (2019) incorporate vulnerability to climatic conditions and natural hazards in Nepal respectively with a wide range of socio-economic factors. However, such studies in Nepal do not include extensive analyses of social vulnerability assessment to earthquakes. The extent of destruction caused by earthquakes depends from one place to another based on the local vulnerability factors such as socio-economic and cultural aspects. For example, the 2015 Gorkha Earthquake damaged more than 700,000 buildings, the majority of

which took place in underdeveloped rural areas with predominant traditional and low-quality masonry houses (Ulak, 2016). In this regard, integrated seismic risk assessment plays a prominent role in determining the areas vulnerable to seismic hazards and reducing the damage in the future. This signifies the need to incorporate the seismic risk assessment with social characteristics. In this study, the country-level earthquake risk estimates from the Global Earthquake Model OpenQuake (Pagani et al., 2014) were analysed by using the input models (seismic hazard sources, fragility functions, and consequence model) given by Chaulagain et al. (2015). The results of earthquake risk estimates were integrated with

vulnerability parameters (social and economic factors) of Nepal. Here, 75 districts and the subdivision, municipalities, and VDCs were considered as the study unit. The administrative unit of Nepal is divided into seven provinces, which are further sub-divided into 75 districts and each district into municipalities and Village Development Committee (VDC). The administrative map of Nepal is shown in Figure 1. This study focuses on social vulnerability and explores the

physical risk from earthquakes at the village and municipal levels.

We assessed the seismic impact potential of the country by moving beyond the physical (direct) impact by integrating physical risk with measures of social vulnerability. And the results are presented in the form of maps along the study area. The main objective is to expand on the information and knowledge of features that are more socially vulnerable to

seismic risks so that policymakers and individuals can carry out a sustainable procedure to reduce the effect in the country. To the author's best knowledge, no previous studies have been documented regarding integrating social vulnerability (preparation of society for any disaster) and seismic risk for Nepal.

**Table 1. Deaths caused by Earthquakes in Nepal (Chaulagain et al., 2018).**

| Year | Magnitude | Death |
| --- | --- | --- |
| 1255 | 7.8 | One-third of Kathmandu Valley Population |
| 1934 | 8.0 | 11,000 |
| 1966 | 6.3 | 24 |
| 1980 | 6.5 | 103 |
| 1988 | 6.9 | 721 |
| 2011 | 6.9 | 111 |
| 2015 | 7.8 | 8,857 |

**2 Theory and background**

**2.1 Social vulnerability variables**

Vulnerability is a multidimensional aspect and it cannot be integrated into a single variable (Cutter and Finch, 2018; Contreras et al., 2020). There are many social vulnerability parameters that determine the impact of natural hazards such as socioeconomic status, geographical features, ethnicity (minority), renter, gender, and age (Burton and Silva, 2016). These intensify the impact of earthquakes; for instance, some people have the privilege to social advantages while some succumb to poverty and discrimination. The households with better economic status can recuperate from disasters better than low-income houses (Mileti, 1999). Similarly, there is lethargic pace in developmental activities, the situation is aggravated by the centralized development only in the capital city, Kathmandu. As a result, people are obliged to work overseas especially in gulf countries for employment opportunities owing to poverty (Aksha et al., 2019). In addition to it, ethnicity also creates barrier in distribution and access to financial resources after disasters (Cutter et al., 2003). Nepal is home to wide number of caste and ethnic identity groups facing deep structural marginalisation (Pherali, 2016). The diverse indigenous communities experience economic, social, and political marginalisation and annexation to infrastructure and financial resources (Mishra et al., 2015). A significant number of minorities, females, and dependent age groups are more vulnerable (Borden et al., 2007). Furthermore, another group of vulnerable populations are renters because, in comparison to the homeowners, renters are financially unprepared for the recovery (Burby et al., 2003). Likewise, the topography of Nepal is also a hindrance in distributing relief materials to the affected regions in time which exacerbates the impact of natural hazards. There are three geographical regions in Nepal: Terai, Hilly, and Himalayan region covering 24%, 42%, and 34% of the total area respectively. The flat, arable, and plain Terai region consists of 50% of total population. The Hilly region is basically situated in the south of Mountain region which is less developed than Terai region. However, the densely populated and highly urbanized cities, Kathmandu and Pokhara lie in Hilly region. The Mountain region is characterized by severe climatic and rugged topographic conditions with limited economic activities and human habitation.

**2.2 Parameters of earthquake risk assessment**

Earthquake risk assessment is the combination of the exposure model, structural vulnerability, and seismic hazard analysis. Generally, the exposure model represents assets like buildings and population (Silva et al., 2020). In this study, the fundamental exposure data include building typologies and district-wise distribution of buildings. Likewise, the structural vulnerability function resembles expected loss at a given ground motion intensity level which can be derived either by empirical methods or by combining fragility and consequence functions (Martins et al., 2021). Using empirical

methods vulnerability is derived in the form of losses from past events at given locations corresponding to the levels of intensity of ground motion. Fragility functions are defined by the probability of exceeding a set of limit states at given intensity measure level (Gomez-Zapata et al., 2021). These functions can be derived empirically or analytically or by modelling the asset behaviour at increasing ground motion levels. Likewise, consequence functions are defined by the probability distribution of loss at a given performance level (Pagani et al., 2014).

The seismic hazard curve describes the annual probability of exceeding a specified ground motion level. The probabilistic approach can be used to derive hazard curves. This approach involves (a) delineation of seismic sources and their characteristics (b) determination of regional seismicity (c) use of appropriate ground motion equation (d) combination of probability in terms of size, location, and ground motion parameter. There are abundant researches on seismic hazard analysis in Nepal which use various source typologies and characteristics. For instance, Stevens et al. (2018) used a mix of fault and area source models — in total six seismic sources with the Gutenberg a and b values along with maximum magnitude estimated for each source zone. Among the sources used in Stevens et. al. (2018), Main Himalayan Thrust is the principle seismic hazard source in Nepal which is a huge, shallow-dipping reverse fault capable of producing largest earthquakes. Similarly, Pandey et al. (2002) divided the whole Nepal region into ten area sources and twenty-four fault sources. Thapa and Guoxin (2013) divided the Nepal region into twenty-three seismic source zones. Chaulagain et al. (2016) also used the same sources to carry out a seismic risk assessment. Similarly, selecting the ground motion prediction equation is one of the important steps in seismic hazard analysis which govern the propagation of seismic ground motion from seismic source to site in terms of magnitude, distance, depth, and other site parameters (Cornell, 1968). However, in the context of Nepal, there are insufficient strong ground motion records to derive a precise equation capturing the actual response spectrum. On top of that, very few research have been conducted in terms of attenuation relationship in Nepal. Previous studies like Chaulagain et al. (2015) and Stevens et al. (2018) have used a combination of GMPEs within the logic tree. The past seismic hazard analyses (Stevens et al., 2018; Chaulagain et al., 2015; Thapa and Guoxin, 2013) have produced a varying range of seismic hazard analyses of Nepal. According to Stevens et al. (2018), in the large part of Nepal, the accelerations in the range of 0.4g-0.6g and 1.0g-3.0g may be expected for 10% and 2% probability of exceedance over 50 years period respectively. Chaulagain (2015) evaluated the estimated peak ground accelerations (PGA in g) at 10% and 2% probability of exceedance in 50 years in the range of 0.22-0.5 and 0.42-0.85g, respectively. Thapa and Guoxin (2013) estimated the PGA (in g) at 10% and 2% probability of exceedance in 50 years in the range of 0.21–0.62 g and 0.38–1.1 g, respectively.

## 3 Materials and methods

This study assesses seismic risk by combining it with the human dimensions within the hazard zone similar to that in Burton and Silva (2016). This approach is an integrated seismic risk assessment. Here we have quantified the social and economic parameters in terms of Social Vulnerability Indices and then integrated those indices with the results of classical probabilistic risk analysis.

### 3.1 Social vulnerability assessment

Social Vulnerability Index (SoVI) method was originally formulated by Cutter et al. (2003) and this index provides a comparative metric depicting an area's relative social vulnerability to hazard. Social vulnerability helps to explain the reason behind the difference in consequences in communities, even though they are subjected to similar levels of ground shaking (Burton and Silva, 2016). We identified the meaningful variables incorporating the socioeconomic and physical context of Nepal. Moreover, to describe the vulnerability at the municipality and VDC level in Nepal, we computed a modified SoVI.

### 3.1.1 Indicators of social vulnerability assessment

For social vulnerability, we extracted data from the most recent national-wise census of Nepal held in 2011 (CBS), Nepal Human Development Report 2014 (UNDP). Table 2 provides the list of all the variables used for social vulnerability
assessment. Out of 45 variables, district-wise indicators were represented by 22 variables and each sub-section of districts (municipality and VDCs) was assumed to have uniform indicator value. Among these 45 variables, seven of them were a weighted combination of multiple variables as shown in Table 2. These weighted variables were obtained from 54 variables mentioned in Table 3. Therefore, altogether 92 variables (45-7+54) were considered for SoVI index. This technique of weighing variables has been used in Principal Component Analysis (PCA) exercised in NHDR (2014).
NHDR (2014) also used the same weightage values for these variables. The modification in the original SoVI is required due to the difference in demographic characteristics between Nepal and the USA and the availability of data. We included variables from various categories like the housing unit status category, which reflects the features of the household, housing characteristics, and facilities. Similarly, population characteristics show female population characteristics, age structure, population density, population growth, child marriage, and disability population.
The cardinality of each indicator (variables) is indicated in Table 2. Positive cardinality (+) means variables have a positive relationship with social vulnerability, while negative cardinality (-) means they have a negative relationship. Each indicator should be normalized to obtain a relatively uniform dimension. Hence, based on cardinality, we used a MINMAX method for each indicator using Eq. (1) and (2), as exercised in Fang et. al. (2019).

For positively related indicators (+), $S_i = (X_i - X_{i,min})/(X_{i,max} - X_{i,min})$         (1)

For negatively related indicators (-), $S_i = (X_{i,max} - X_i)/(X_{i,max} - X_{i,min})$         (2)

Where, $X_i$ is the original value of indicator i; $X_{i,max}$ and $X_{i,min}$ are the maximum and minimum values of the variable $X_i$. $S_i$ is the standard value of index i, which is in the range of 0 and 1.

**Table 2. Variables used to construct social vulnerability index and their loading values after PCA.**

| S.N. | Description | Category | Data Source | Cardinality | Loadings |
|---|---|---|---|---|---|
| 1 | Percentage of households that owned a house | Housing Unit Status | a | - | -0.502 |
| 2 | Weighted Foundation Index per household ** | Housing Unit Status | a | - | 0.863 |
| 3 | Weighted Wall Index per household ** | Housing Unit Status | a | - | 0.872 |
| 4 | Weighted Roof Index per household ** | Housing Unit Status | a | - | -0.645 |
| 5 | Weighted Drinking Water Index per household ** | Housing Unit Status | a | - | 0.498 |
| 6 | Weighted Cooking Index per household ** | Housing Unit Status | a | - | 0.673 |
| 7 | Weighted Electricity Index per household ** | Housing Unit Status | a | - | 0.72 |
| 8 | Percentage of households without toilet facility | Housing Unit Status | a | + | 0.533 |
| 9 | Percentage of households without any of following facilities: radio, television, mobile, refrigerator, vehicles, internet | Housing Unit Status | a | + | 0.653 |
| 10 | Percentage of households with radio facilities | Housing Unit Status | a | - | 0.395 |
| 11 | Percentage of households with television facilities | Housing Unit Status | a | - | 0.711 |
| 12 | Percentage of households with internet facilities | Housing Unit Status | a | - | 0.614 |

| | | | | | |
|---|---|---|---|---|---|
| 13 | Percentage of households with vehicles | Housing Unit Status | a | - | -0.51 |
| 14 | Percentage of absentee households | Housing Unit Status | a | + | -0.885 |
| 15 | Average household size | Housing Unit Status | a | + | 0.511 |
| 16 | Percentage of households with child as household head * | Housing Unit Status | a | + | 0.546 |
| 17 | Percentage of households with female as household head * | Housing Unit Status | a | + | -0.797 |
| 18 | Percentage of household with 5+ members * | Housing Unit Status | a | + | 0.666 |
| 19 | Housing Density * | Housing Unit Status | a | + | -0.914 |
| 20 | Percentage of population that is females | Population | a | + | -0.84 |
| 21 | Percentage of children under 5 years | Population | a | + | 0.748 |
| 22 | Percentage of children aged 5 to 14 | Population | a | + | 0.705 |
| 23 | Percentage of people aged 30 to 49 | Population | a | - | 0.712 |
| 24 | Percentage of elder population (65+) | Population | a | + | -0.483 |
| 25 | Percentage of population with disabilities (blind, deaf, mental) | Population | a | + | -0.374 |
| 26 | Percentage of child marriages * | Population | a | + | 0.745 |
| 27 | Population Growth (2001 - 2011) * | Population | b | - | -0.846 |
| 28 | Net Migration Rate * | Population | b | - | -0.85 |
| 29 | Population Density * | Population | a | + | -0.89 |
| 30 | Population per each hospital and PHCC/HCC * | Health | e | + | 0.51 |
| 31 | Population per each health posts and sub-health post * | Health | e | + | 0.773 |
| 32 | Life Expectancy * | Health | b | - | 0.776 |
| 33 | Infant Mortality Rate (Per 1000 Birth) * | Health | b | + | 0.706 |
| 34 | Literacy Rate | Education | a | - | 0.469 |
| 35 | Weighted Education Level Index per capita ** | Education | a | - | 0.717 |
| 36 | Population per each school * | Education | g | + | 0.526 |
| 37 | Human Poverty Index * | Economy | d | + | 0.803 |
| 38 | Human Development Index (2011) * | Economy | d | - | 0.878 |
| 39 | Budget allocation per capita * | Economy | f | - | 0.835 |
| 40 | Per capita income, Rs. at market price * | Economy | d | - | 0.751 |
| 41 | Percentage of population that are economically active * | Economy | d | - | 0.488 |
| 42 | Gross Domestic Product (Value Added) Rs. In Million (per Capita) * | Economy | d | - | 0.739 |
| 43 | Labour Productivity per capita * | Economy | d | - | 0.871 |
| 44 | Population per each small industry * | Economy | c | + | 0.429 |
| 45 | Percentage of employment that are female * | Economy | a | - | 0.554 |

a   National Population and Housing Census 2011 (CBS (2012))
      b   Population Monologue V01 (CBS (2014b))
      c   Population Monologue V03 (CBS (2014a))
      d   Nepal Human Development Report (Sharma et al., 2014)
      e   Department of Health Services (2013)
f   Budget report for year 2070-71 (2013 - 14)
      g   Department of Education (2013 – 14)
      *   District-wise data
      ** Weighted index calculated as per Table 3


**Table 3. Weights corresponding to the weighted variables, as defined in Table 2.**

| Variables | Weightage | Variables | Weightage |
|---|---|---|---|
| **A. Weighted Foundation Index** Types of foundation in houses | | **E. Weighted Cooking Index** Main cooking fuel | |
| RCC with pillar | 5 | LP gas | 6 |
| Cement bonded bricks/stone | 4 | Electricity | 6 |
| Mud bonded bricks/stone | 3 | Kerosene | 5 |
| Wooden pillar | 2 | Bio gas | 4 |
| Others | 1 | Wood/firewood | 3 |
| Not Stated | 1 | Santhi/guitha (cow dung) | 2 |
| **B. Weighted Wall Index** Types of walls in houses | | Others | 1 |
| Cement bonded bricks/stone | 6 | Not Stated | 1 |
| Mud bonded bricks/stone | 5 | **F. Weighted Electricity Index** Main source of light | |
| Wood/ planks | 4 | Electricity | 5 |
| Bamboo | 3 | Solar | 4 |
| Unbaked brick | 2 | Bio gas | 3 |
| Others | 1 | Kerosene | 2 |
| Not Stated | 1 | Others | 1 |
| **C. Weighted Roof Index** Types of roofs in houses | | Not Stated | 1 |
| RCC | 7 | **G. Weighted Education Index** Highest level of education of each individual | |
| Tile/slate | 6 | Post Graduate equiv.  and above | 9 |
| Galvanized iron | 5 | Graduate and equiv. | 8 |
| Wood/planks | 4 | Intermediate and equiv. | 7 |
| Mud | 3 | S.L.C. and equiv. | 6 |
| Thatch/straw | 2 | Secondary (9 -10) | 5 |
| Others | 1 | Lower secondary (6 -8) | 4 |
| Not Stated | 1 | Primary (1-5) | 3 |
| **D. Weighted Drinking Water Index** Main source of drinking water | | Beginner | 2 |
| Tap/piped water | 7 | Others | 1 |
| Covered well/kuwa | 6 | Non-Formal | 1 |
| Tubewell/handpump | 5 | Not Stated | 1 |
| Uncovered well/kuwa | 4 | | |
| Spout water | 3 | | |
| River/stream | 2 | | |
| Others | 1 | | |
| Not Stated | 1 | | |

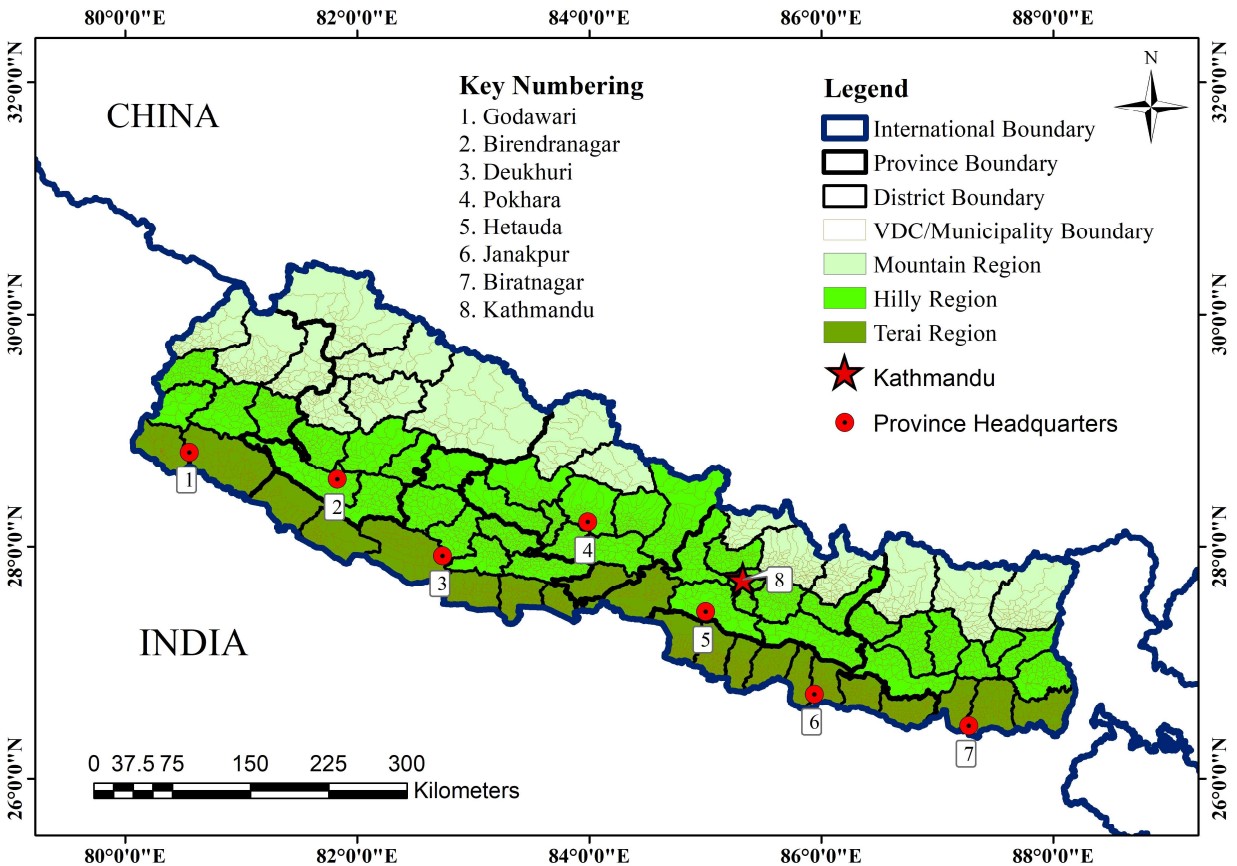

**Figure 1. Administrative Map of Nepal showing VDCs, municipalities, 75 districts, three geographical regions, provinces, and province headquarters.**

### 3.1.2 Calculation of Social Vulnerability Index (SoVI) by Principal Component Analysis (PCA)

Social Vulnerability Index was evaluated by incorporating socio-economic variables through a mathematical procedure called Principal Component Analysis (PCA). PCA transforms a number of possibly correlated variables into a smaller

number of uncorrelated components (Abdi and Williams, 2010). The main idea of PCA is to reduce the dimensionality of a dataset with a large number of inter-related indicators, whilst retaining the maximum possible variation present in the data set (Jolliffe, 2002). The procedure of Principal Component Analysis used in this study is mentioned in further sections.

### 3.1.2.1 Number of principal components

It is very crucial to determine the number of components to carry out PCA (Franklin, 1995). We used Parallel Analysis (PA) by Horn (1965). Various studies like Humphreys and Montanelli Jr. (1975), Zwick and Velicer (1986), and Thompson and Daniel (1996) have shown that PA is an appropriate method to determine the number of factors. These studies assert that this method (PA) is the best available alternative to calculate the number of factors to be retained. In this method, eigenvalues from PCA prior to rotation were compared with 'expected' eigenvalues which were obtained

by simulating normal random samples with identical dimensionality (same number of samples and variables) using a Monte-Carlo simulation process. Initially, a factor was considered significant if the associated eigenvalue was bigger than the mean of those obtained from the random uncorrelated data. The default (and recommended) values for a number of random correlation matrices and percentile of eigenvalues are 100 and 95, respectively (Cota et al., 1993; Glorfeld, 1995; Velicer et al., 2000). We used parallel analysis engine developed by Vivek et al. (2017) to calculate corresponding

random eigenvalues. From parallel analysis, there were eight components with larger associated eigenvalues than that

from the Monte-Carlo simulation as shown in Table 4. These eight components explained 77.51% of the variance in all variables.

We also used two rules-of-thumb to calculate the number of components to be retained for comparison. The first rule of thumb used in this study was proposed by Kaiser (1960). As per this rule, only those principal components with eigenvalues greater than 1.0 were retained. As seen in Table 4, just like parallel analysis, Kaiser's rule also indicated eight principal components. Cattell scree test was also used as second rule of thumb and the test proposed in Cattell (1966) is based on the scree-plot (Eigenvalues vs the number of components). According to this test, a point where the scree plot moves from steep to shallow was taken as a cutting-off point as shown in Figure 2 which also indicated eight principal components similar to parallel analysis.

Table 4. Initial Eigenvalues, variances, and results of Parallel analysis for first ten principal components.

| Component | Initial Eigenvalues (a) | % Of Variance | Cumulative % | 95% Percentile Eigenvalue (Parallel Analysis) (b) | Parallel Analysis: Remarks |
|---|---|---|---|---|---|
| 1 | 13.215 | 29.366 | 29.366 | 1.227612 | a > b |
| 2 | 9.201 | 20.447 | 49.813 | 1.199826 | a > b |
| 3 | 3.541 | 7.87 | 57.683 | 1.183024 | a > b |
| 4 | 3.096 | 6.88 | 64.563 | 1.170289 | a > b |
| 5 | 1.787 | 3.972 | 68.535 | 1.160148 | a > b |
| 6 | 1.488 | 3.308 | 71.842 | 1.147872 | a > b |
| 7 | 1.345 | 2.99 | 74.832 | 1.136596 | a > b |
| 8 | 1.206 | 2.681 | 77.513 | 1.126383 | a > b |
| 9 | **0.929** | 2.064 | 79.577 | **1.11708** | **a < b** |
| 10 | 0.79 | 1.755 | 81.332 | 1.107092 | |

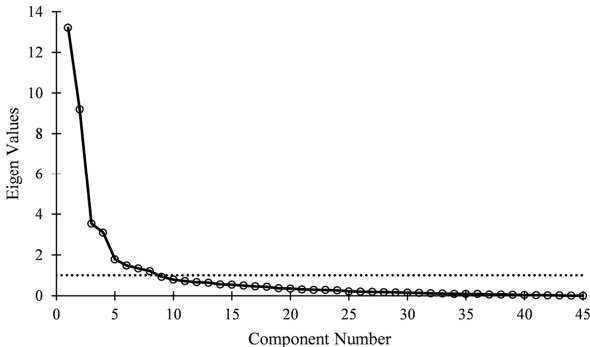

Figure 2. Scree plot (Eigen Values Vs Components).

**3.1.2.2 Suitability of data for PCA**

We performed two tests Kaiser-Meyer-Olkin test and Bartlett's test to check the adequacy of data for PCA. The Kaiser-Meyer-Olkin (KMO) measure of sampling adequacy depicts the proportion of variance in the variables that might be caused by underlying factors (Kaiser, 1970; Fekete, 2009). KMO-value greater than 0.8 was considered good, while KMO-value less than 0.5 required some remedy, either by deleting or adding variables (IBM Support, 2020). Similarly, Bartlett's test of sphericity is the suitability test, where the value below 0.05 indicates the variables are related and

suitable for structure detection. In this study, KMO value of 0.873 and Bartlett's test value of 0.000 passed the requirements of data for PCA.

### 3.1.2.3 Statistical Analysis

PCA was carried out in Statistical Package for Social Science (SPSS version 21.0). We employed Varimax rotation with Kaiser normalization as applied by Aksha et al. (2019), Fekete (2009) which maximized the variance shared among data and eases the interpretation by rotating the axes of the components perpendicular to them. For the interpretation of the result, we suppressed the absolute loading value less than 0.30 and considered eigenvalues greater than 1.0 as in Fekete (2009). Due to the lack of justifiable method and evidence for weighting components, an equal weighting, and additive approach was considered as exercised in similar studies Cutter et al. (2003) and Aksha et al. (2019). The loadings after PCA are presented in Table 2. Thereafter, SoVI scores were calculated by summing the scores of all principal components. As presented in the paper Tate (2012), SoVI scores were used in the form of standard deviations (z-scores) or quintiles to emphasize their relative value. Furthermore, z-scores were considered to classify the social vulnerability of each VDC and municipalities into five groups and then plotted the results in map form using ArcGIS.

### 3.2 Assessment of physical risk

The Classical Probabilistic Seismic Hazard Analysis (PSHA) based risk calculator was performed to calculate the annual average loss using OpenQuake. This calculator combines numerical integration, physical vulnerability functions of the assets, and seismic hazard curve at the location to calculate the loss distribution for the asset within a specified time period (Pagani et al., 2014). The calculator requires an exposure model describing the distribution of building typologies, physical vulnerability functions for each building type, and hazard curves calculated in the region of interest. The hazard curves required were also calculated using the OpenQuake engine using the classical PSHA approach. For hazard curves derivation, a source model and ground motion prediction were inputted in the OpenQuake engine. Finally, the value of annual average loss (AAL) for each VDCs and municipalities was rescaled into the range between 0 and 1 using MIN-MAX rescaling (Equation 1).

### 3.2.1 Source model

In this study, the twenty-three source zones similar to that of Thapa and Guoxin (2013) were considered for probabilistic seismic hazard analysis. The seismic source zones are shown in Figure 3. The delineated sources were assumed to be homogenous in terms of their seismicity such that every point was assigned an equal probability of occurrence of an earthquake. Thapa and Guoxin (2013) determined 'b' value of 0.85 for the entire region. Here, we considered the same 'b' value as proposed in that study. Generally, small-magnitude earthquakes have a minute effect on infrastructures. Therefore, for the hazard analysis, the minimum magnitude ($M_w$) within all source zone was considered 4.0. Similarly, the hypocenter depth of 10 km was used for the entire region.

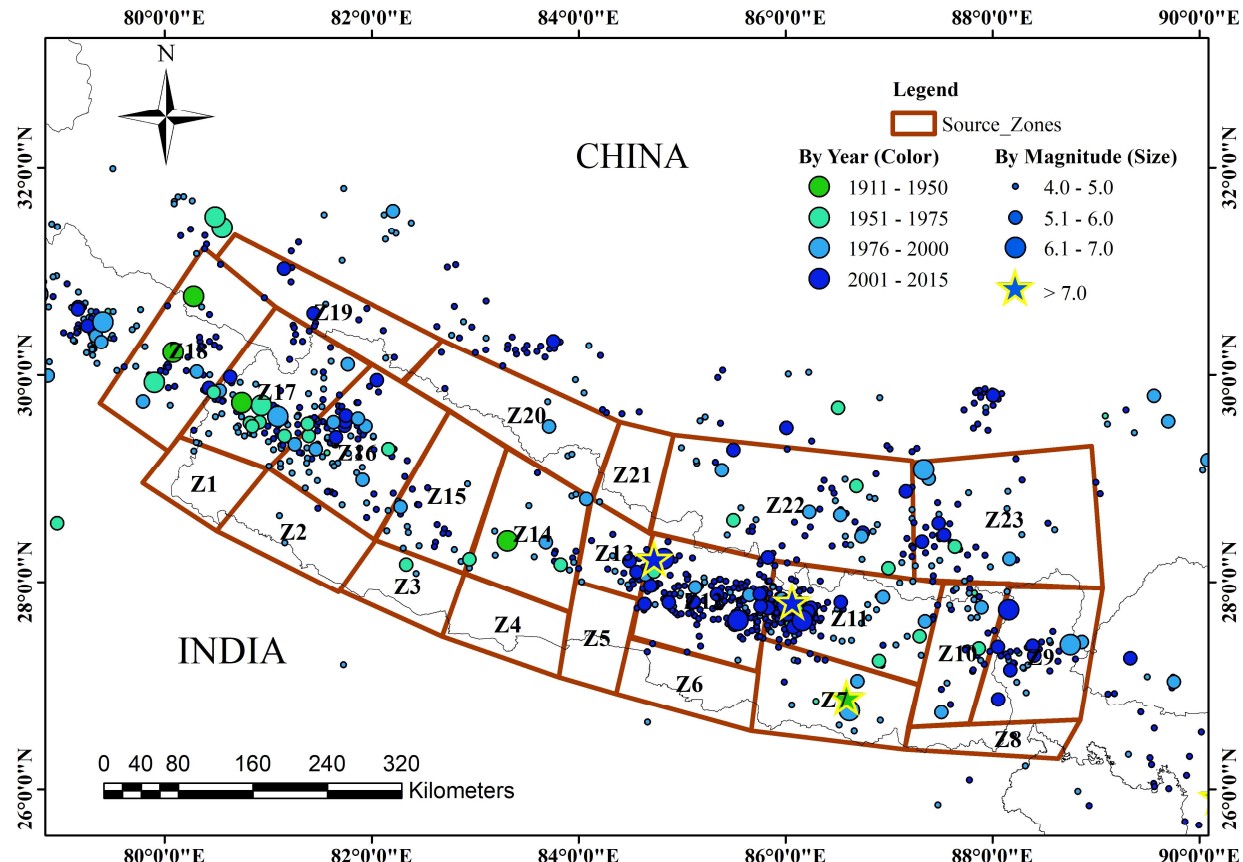

**Figure 3. Seismic Source Zones of Nepal (Thapa and Guoxin, 2013).**

### 3.2.2 Attenuation relationship (Selection of ground motion prediction equation)

We assumed the tectonic region as a shallow crust and subduction interface like that in Chaulagain et al.(2016). Atkinson and Boore (2003), Youngs et al. (1997), Campbell and Bozorgnia (2008), Chiou and Youngs (2008), and Boore and Atkinson (2008) were used. These equations were used within a logic tree (equal weights for each equation) to conduct probabilistic seismic hazard and risk analysis in OpenQuake.

### 3.2.3 Exposure model and physical vulnerability model

In this study, the building description and data from Census 2011 were used to develop the exposure model without considering the industrial or commercial buildings. In other words, only residential buildings were considered for the exposure model. The total number of households according to Census 2011 is 5,423,297. The exposure models used in the study are part of seismic risk assessment with uncertainties, although present studies like Kalakonas et al. (2020), Bal et al. (2010), and Gomez Zapata et al. (2022) have pointed out how the epistemic uncertainties embedded in exposure
models are propagated throughout the computation of seismic risk.

 We considered five types of buildings — mud bonded, cement-bonded, reinforced cement concrete (RCC), wooden, and adobe. Most of the residential building stock in Nepal consists of mud-mortar/bonded brick masonry buildings. In remote areas, wooden buildings are abundant whereas, in the central region, especially in Kathmandu and urban areas, cement bonded or reinforced concrete buildings are present. The area and construction cost of each building type is
shown in Table 5 as considered by Chaulagain et al. (2015). The spatial distribution of total buildings across the country is shown in Figure 4a and the individual building typology is summarized in Box and Whisker plot as shown in Figure

4b. The average values of RCC with pillar, Mud-Bonded, Cement-Bonded, Wooden-pillar, and Adobe are 135.73, 603.36, 239.91, 340, and 46.33 respectively as presented in Figure 4b.

On the other hand, the average annual loss was evaluated using OpenQuake. In this study, the fragility model developed by Chaulagain et al. (2015) was adopted for different building types. To define fragility functions in a discrete manner, for each limit state, a list of intensity measure levels and their corresponding probabilities of exceedance must be provided. The intensity measure level in terms of peak ground acceleration (PGA (g)) was used. The fragility curves for different building typologies at each limit state are shown in Figures 5 and 6. After defining fragility functions, it is also important to assess the correlation between the logarithmic means and standard deviations of each limit state, which are represented by μ and σ, respectively as shown in Table 6. Thereafter, the fragility curve was inputted in the Vulnerability Modeller's Toolkit (VMTK) developed by GEM OpenQuake to derive the physical vulnerability model. VMTK is a framework divided into six modules which can be used to derive fragility function via non-linear dynamic analysis and also the physical vulnerability function using fragility model and consequence model (Martins et al., 2021). In this process, the fractions of buildings in each damage state were multiplied by the associated damage ratio (from the consequence model), in order to obtain a distribution of loss ratio for each intensity measure type (Pagani et al., 2014).The damage ratio 0.3, 0.6, and 1.0 were used for each damage type: moderate, extensive, and collapse, respectively as per Chaulagain et al. (2015). The vulnerability curves for each building typologies used in this study are shown in Figure 7.

**Table 5. Area and construction cost of different Building Type (Chaulagain et al., 2015)**

| Building Type | Area per building (m$^2$) | Construction cost (€/m$^2$) |
|---|---|---|
| Adobe | 60 | 150 |
| Mud Bonded | 70 | 225 |
| Cement Bonded | 80 | 275 |
| Wooden | 60 | 200 |
| RCC | 80 | 325 |

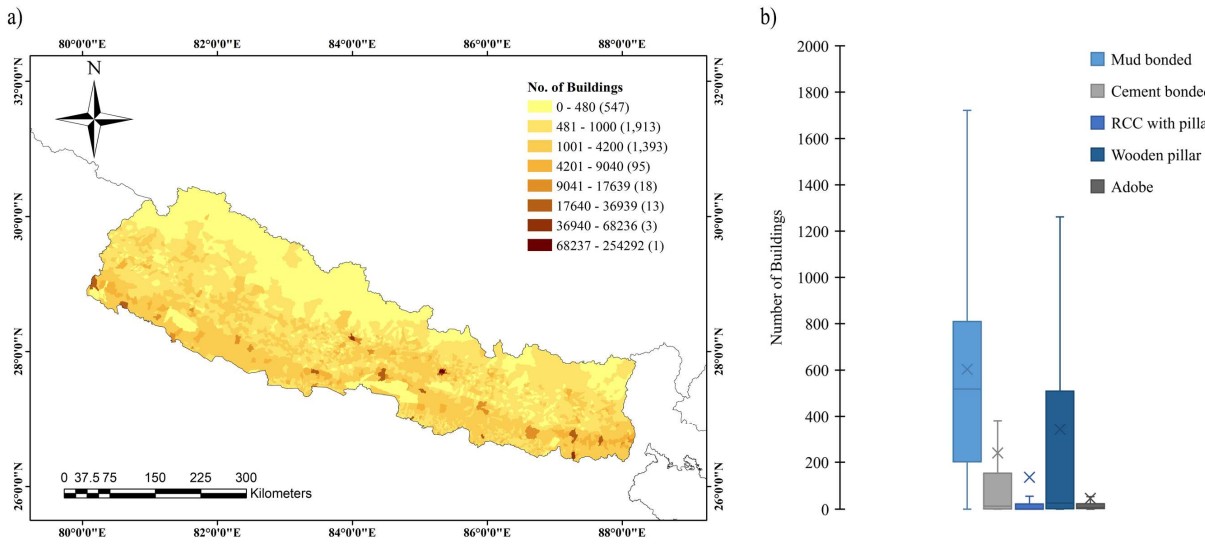

**Figure 4. a) Spatial distribution of total buildings in Nepal b) Box and Whisker plot describing the distribution of each building types**

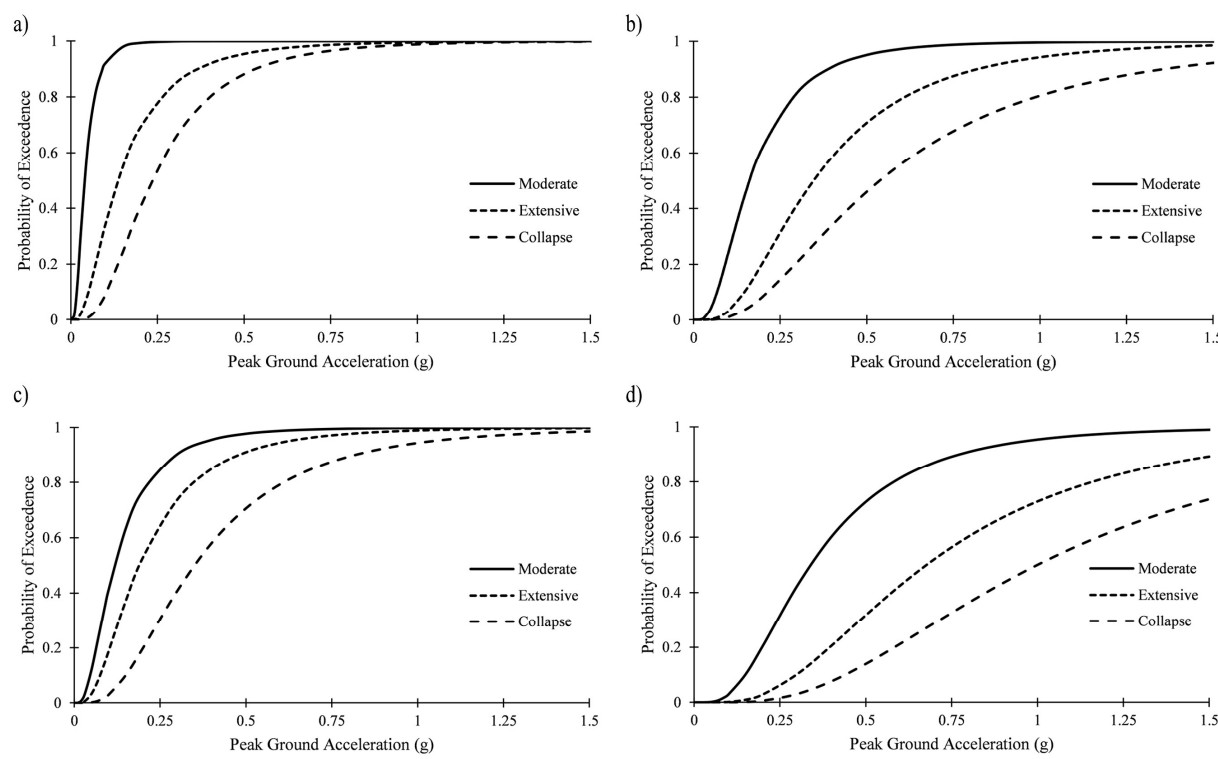

**Figure 5. Fragility curves for a) adobe b) cement bonded c) mud mortar d) wooden buildings**

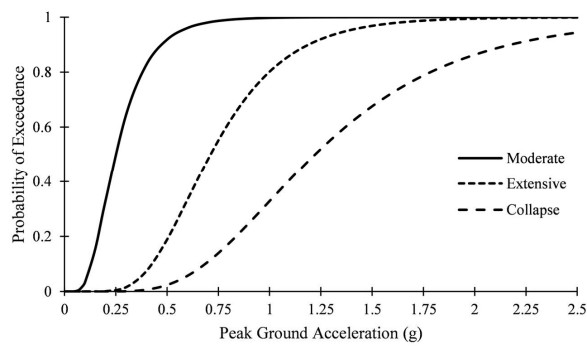

**Figure 6. Fragility curves for RCC buildings**

**Table 6. Mean and standard deviation per damage state for each building type (Chaulagain et al., 2015).**

| Building type | Moderate Damage | | Extensive Damage | | Collapse | |
|---|---|---|---|---|---|---|
| | μ | σ | μ | σ | μ | σ |
| Adobe | -3.22 | 0.65 | -1.99 | 0.77 | -1.45 | 0.64 |
| Mud Bonded | -2.14 | 0.72 | -1.66 | 0.72 | -1.05 | 0.66 |
| Cement Bonded | -1.82 | 0.68 | -1.06 | 0.67 | -0.62 | 0.72 |
| Wooden | -1.08 | 0.64 | -0.39 | 0.64 | 0.00 | 0.64 |
| RCC | 0.35 | 0.17 | 0.85 | 0.2 | 1.35 | 0.32 |

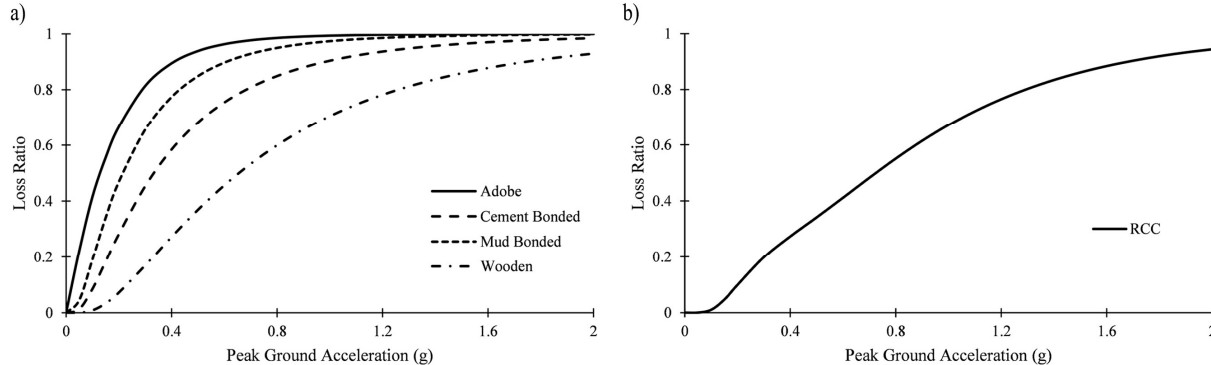

**Figure 7. Vulnerability curves a) adobe, cement bonded, mud bonded, and wooden buildings b) RCC buildings**

### 3.3 Integrated risk assessment

An integrated risk index was constructed by combining the social vulnerability index and estimates of average annual loss in rescaled metrics. The framework or workflow of the integrated risk assessment is shown in Figure 8. The first step in Figure 8, seismic hazard, was evaluated using the Probabilistic approach. The geographic features represent exposure modelling for residential buildings and their physical vulnerability. By combining these parameters, seismic risks were evaluated in terms of Average Annual Loss (AAL) which was further recomputed by using the Min-Max

rescaling method. The physical vulnerability and exposure model interact with the social and economic parameters or overall capacity of the population to sustain hazards (Burton and Silva, 2016). The social features define socio-economic parameters related to the demographic population to prepare for, react to, and recuperate from damaging events (Burton and Silva, 2016). The integrated risk is the combination of physical risk and a set of contextual and social vulnerability conditions (Carreño et al., 2012). In this regard, the integrated risk was evaluated grounded on direct factors or physical

risk and socio-economic factors. The integrated risk index (RT) was calculated using Eq. (3):

$$R_T = R_f \left(1 + F\right) \tag{3}$$

The Moncho's equation (Eq (3)) was used to evaluate convoluted risk, where $R_f$ is a physical risk index or average annual loss estimate, and F is a social fragility index or aggravating coefficient (Glorfeld, 1995). This technique and its successful application can be found in numerous studies due to its simplicity and successful applications (Burton and

Silva, 2016; Carreño et al., 2012; Khazai, Merz, et al., 2013; Fernandez et al., 2006). The calculated integrated risk was evaluated using the OpenQuake integrated risk modelling toolkit. The integrated risk modelling toolkit (IRMT) is a plugin developed by GEM Foundation and available in QGIS opensource platform that allows to build a composite framework to assess physical risk and social characteristics that affect the earthquake risk. The diagrammatic workflow of social and physical risk indicators developed by OpenQuake IMRT is shown in Figure 9. The integrated toolkit

involves details from selection of indicators to the detailing and mapping of composite risk assessment.

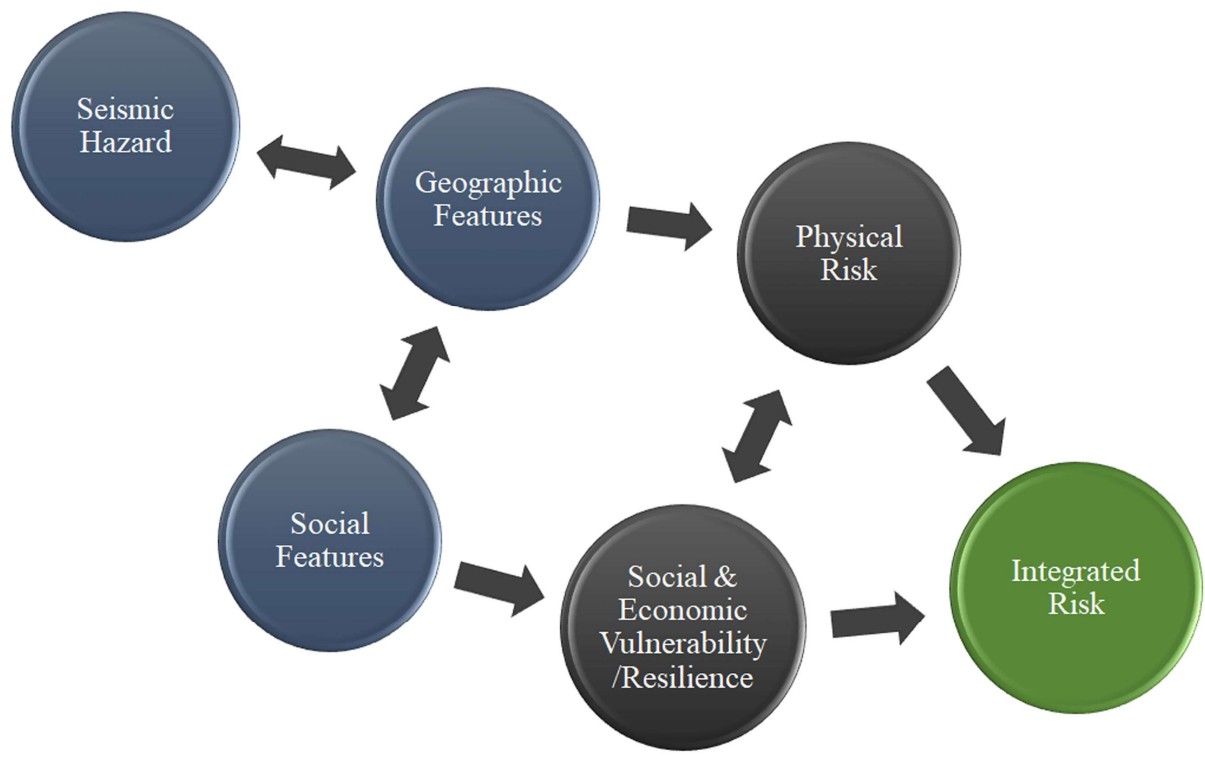

**Figure 8: Framework for Integrated Risk Approach (Burton and Silva, 2016)**

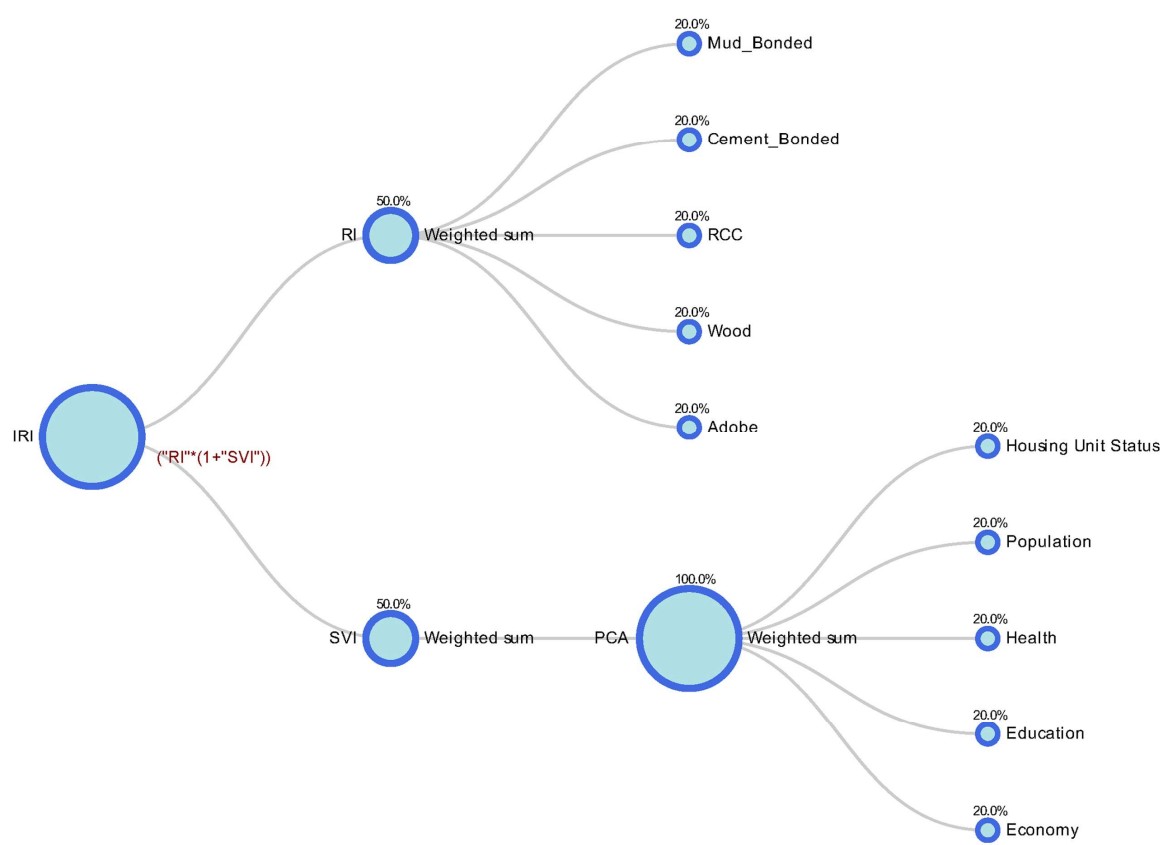

**Figure 9. Workflow showing social and physical risk indicators in QGIS IRMT.**

# 4 Results

## 4.1 Results of social vulnerability analysis

The results of social vulnerability analysis are shown in Figure 10 which depicts the distribution of total SoVI scores across the country. SoVI scores for each district provide relative comparison within the districts' sub-division units. The districts mapped with darker red shades demonstrate higher rates of social vulnerability. It can be seen from Figure 10 that there is a higher degree of risk in the Far-Western region hosting the places like Godawari and Deukhuri. This region is one of the most rural areas of Nepal with nominal availability of infrastructure. The results also illustrate that the Terai region with cities like Janakpur and Biratnagar show a higher level of social vulnerability. These cities may be at greater risk as a result of infrastructure exposure. On the other hand, the central and eastern regions of Nepal with cities Kathmandu, Pokhara, and Hetuada are comparatively at lower risk. The social vulnerability map of the country demonstrates higher levels of social vulnerability in rural areas with few exposed assets and industrial areas with exposure of infrastructure whereas the lowest levels of social vulnerability in urban and populated cities like Kathmandu, Pokhara, and Hetauda.

To further explore the overall social vulnerability observed in Figure 10, we generated the maps of sub-components: housing unit, population, health, education, and the economy as shown in Figure 11. The first Figure 11a) shows that highly vulnerable areas under the housing unit category are concentrated in Far-Western Hill and Eastern Terai regions. The houses in rural areas are highly vulnerable due to their old age and lack of amenities like retrofitting and repairing. The population component (Figure 11b) exhibits a high level of vulnerability in Kathmandu, Janakpur, and some parts of the Far-Western region. There is a dense population and unorganized urbanization in Kathmandu, whereas the Far-Western region hosts minorities and an old age population. Under the health component (Figure 11c), there is an intense degree of vulnerability in the Terai region and the Hilly parts of the Far-Western region. These regions have the least access to health facilities. Similarly, the component (Figure 11d), education, reveals great vulnerability in the Eastern Terai region. The component economy (Figure 11e) shows a very high degree of vulnerability in the mid-part of the Far-Western region and Terai region. There is a high level of vulnerability in the Far-Western region of the country due to the prevalence of a higher number of minorities, illiteracy rate, nominal infrastructural access, and poor economic status.

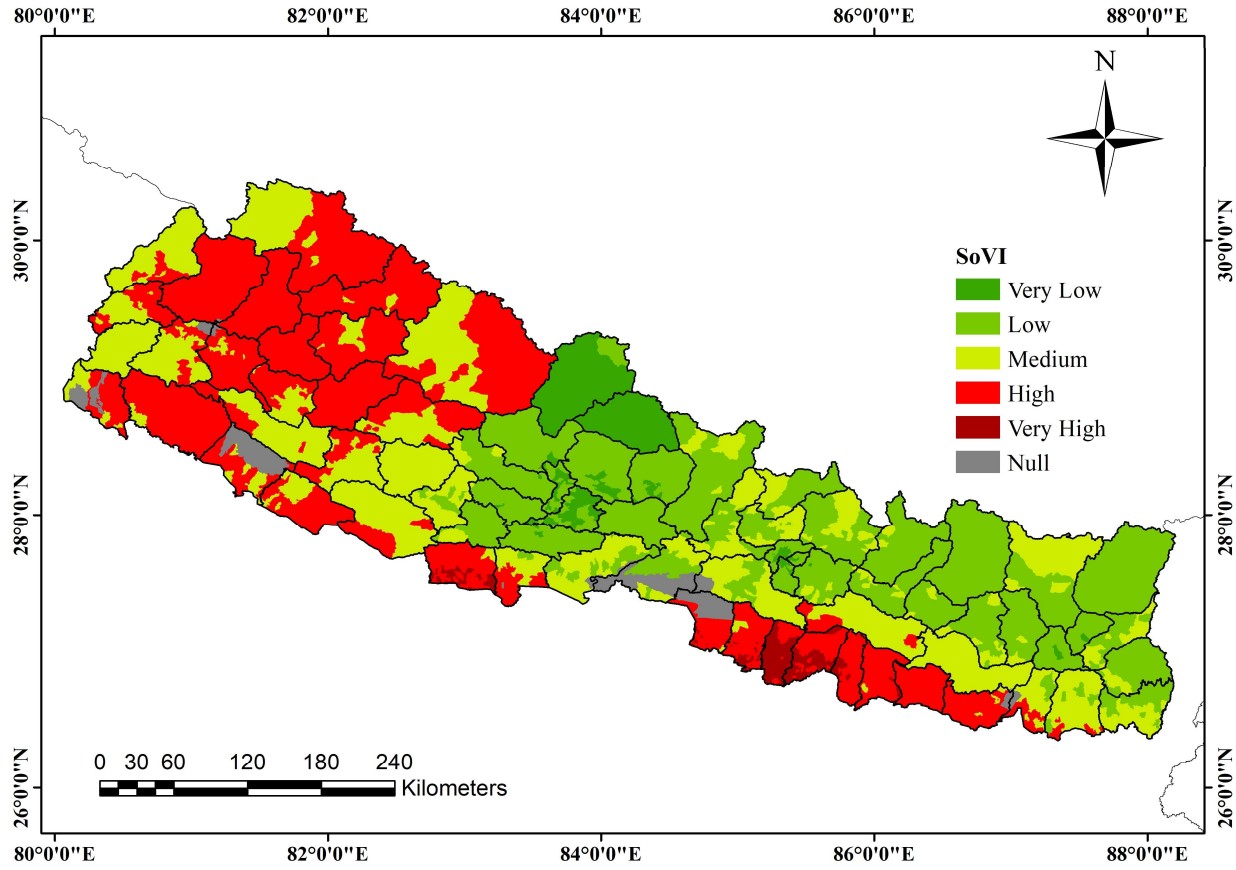

**Figure 10: Spatial distribution of Social Vulnerability Index in districts of Nepal.**

365

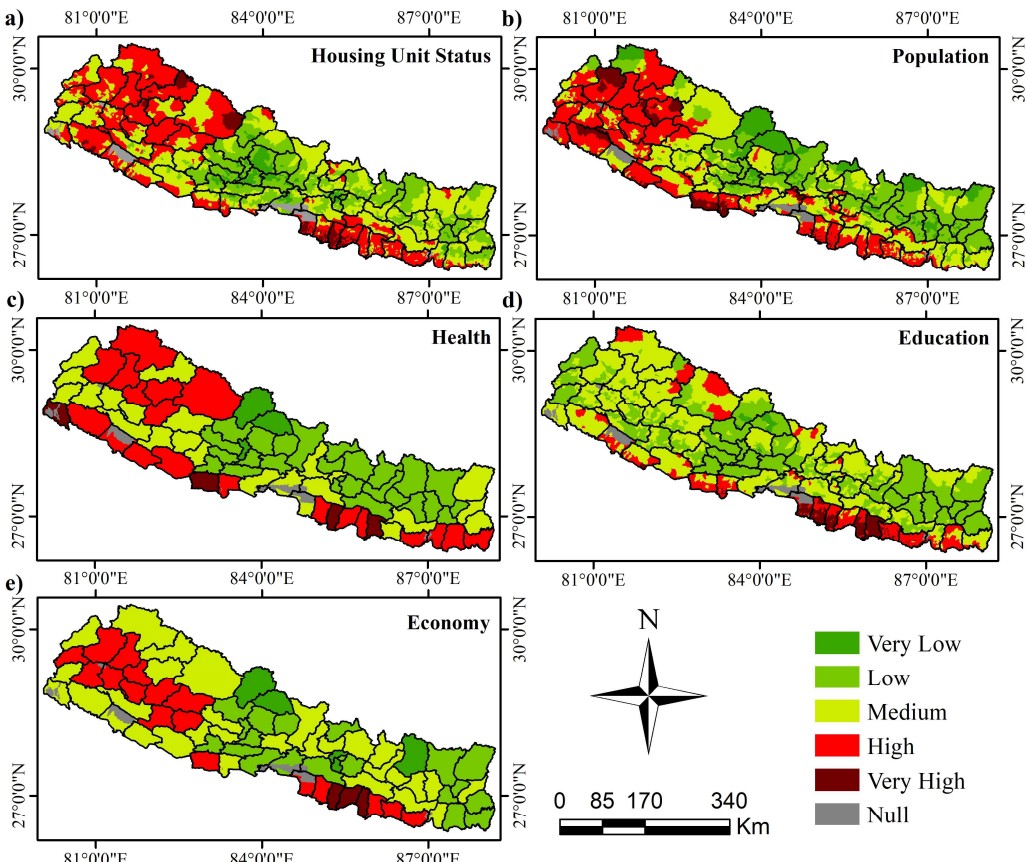

**Figure 11. Spatial distribution of social vulnerability of sub-categories.**

**4.2 Results of seismic risk assessment**

The probabilistic seismic hazard analysis was carried out and then the physical risk estimations were made as shown in Figure 12. The physical risk in terms of Average Annual Loss (AAL) index obtained from the risk analysis was classified into five quintiles from very low ($<$ -1.5 standard deviation) to very high ($>$ 1.5 standard deviations) vulnerability. Figure 12 shows the distribution of AAL per capita in monetary terms and Figure 13 shows the distribution of Seismic Risk index across the country. The null region in the maps represent the areas which are national parks and wildlife reserves with nominal population. From Figure 13, it is observed that the Terai region, especially the eastern terai and central terai lie in higher seismic risk category. Kathmandu Valley also lies in a very high-risk category. Contrary to social vulnerability, the western part of Nepal lies in the lower AAL value region.

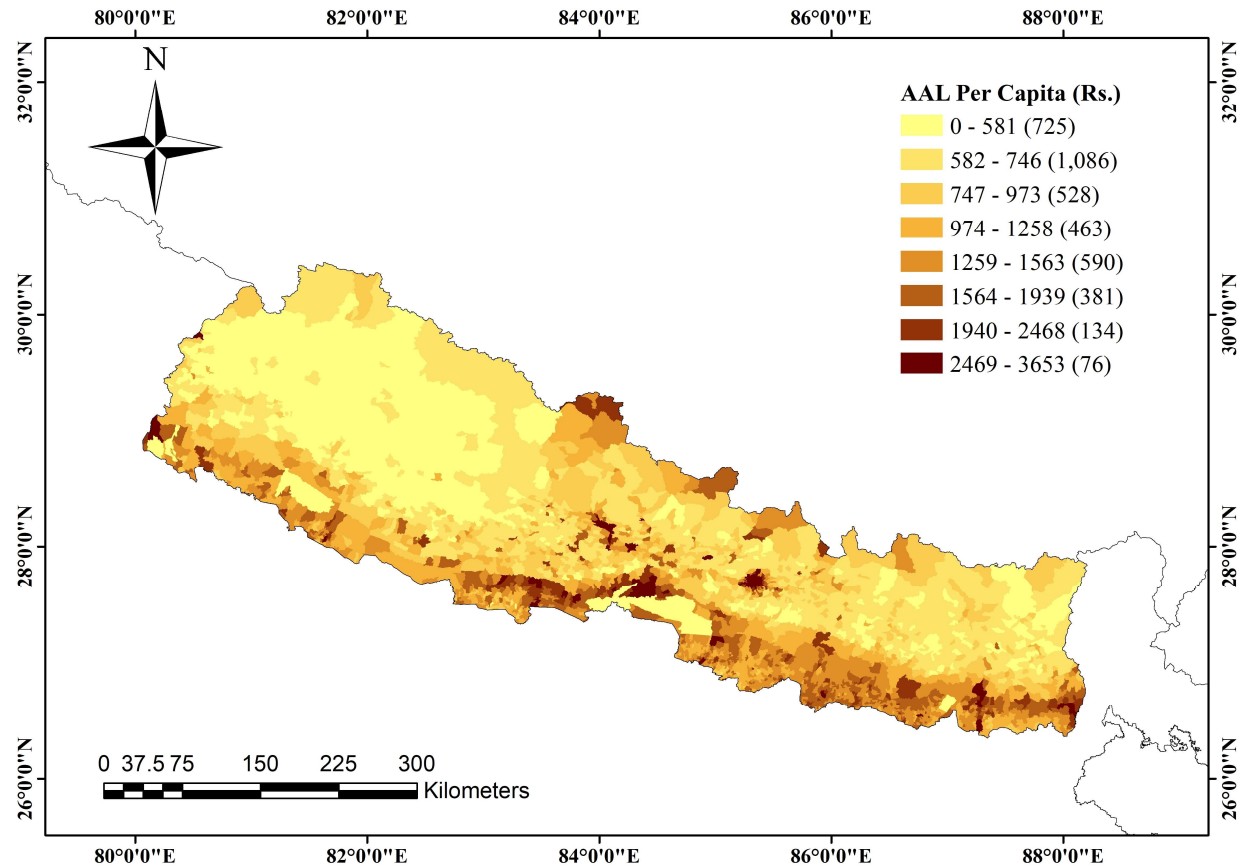

380        **Figure 12. Average Annual Loss per capita, as calculated from OpenQuake, for Nepal.**

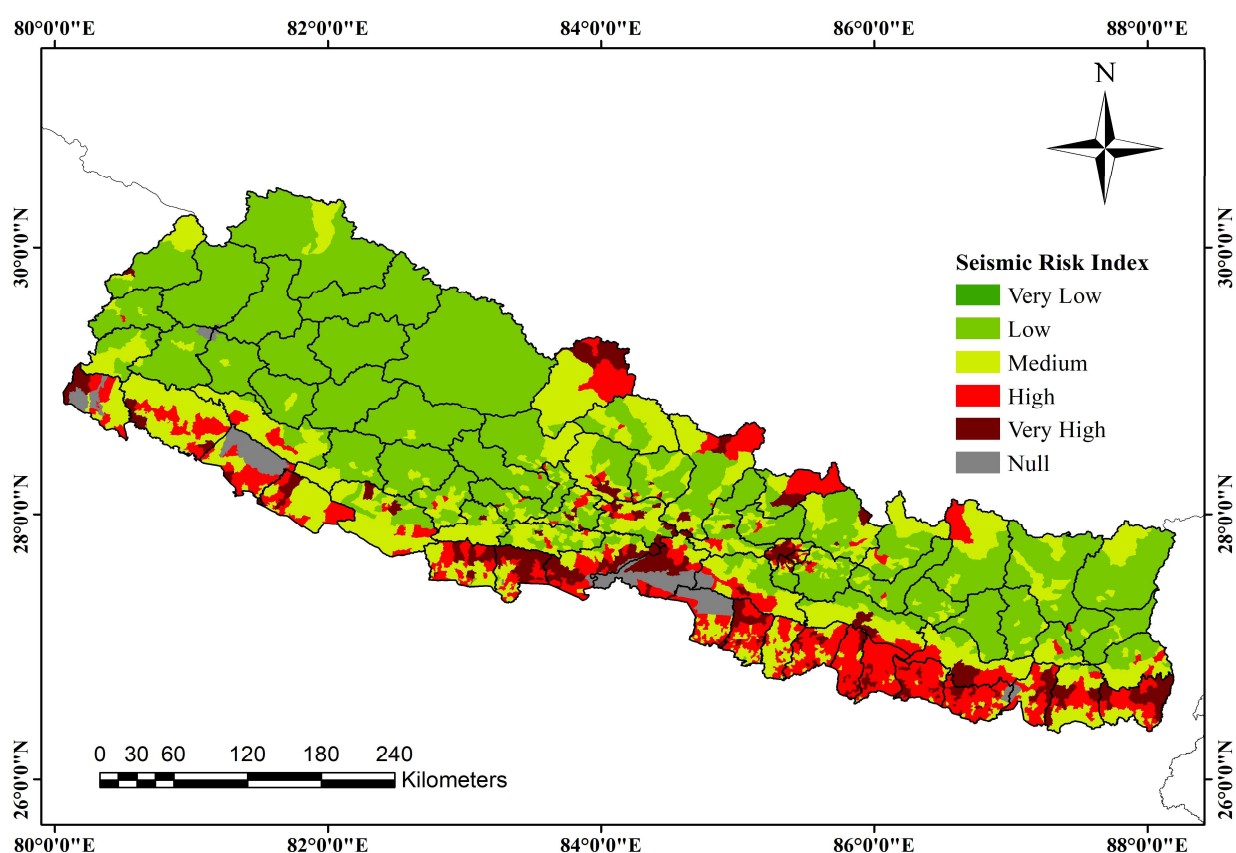

**Figure 13. Spatial distribution of Seismic Risk Index in Nepal.**

**4.3 Results of integrated risk assessment**

Social vulnerability shows the intensity of the impact of any disaster. When combined with the impact of seismic action, the true nature of the distribution of seismic risk becomes evident. As shown in Figure 14, the integrated risk is higher in the Terai region in cities like Janakpur and Biratnagar. Kathmandu region has a low SoVI index, but a high integrated risk index. Similarly, the Far-Western Hills and Mountain regions are found to be in the low risky region, even though they have a high SoVI index. However, due to their high social vulnerability, these regions should still be depicted as a concern. Despite having a low number of houses, the houses may be of lower quality, which is inclined to suffer damage to even low-magnitude earthquakes, and these regions may not have enough resources for mitigation measures. On top of that, even though they are in a low seismic risk region, the respective population may be at high risk to other disasters like a landslide, flood, glacier, and other weather conditions.

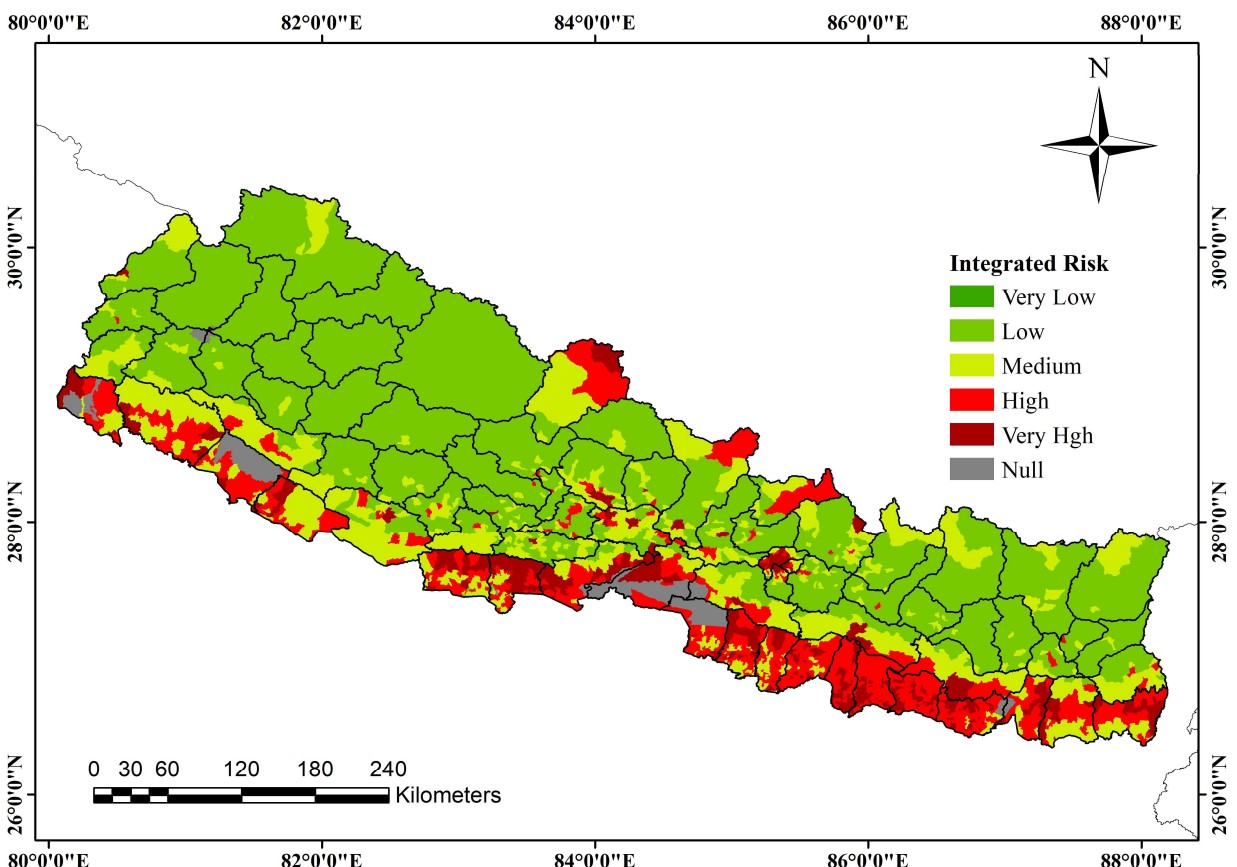

**Figure 14. Spatial distribution of Integrated Risk Index in districts of Nepal.**

**5 Discussion**

The main purpose of this study is to assess the integrated seismic risk by combining the socio-economic parameters and physical risk. The methodology by Cutter et al. (2003) was used to compute social vulnerability components. The total SoVI scores were calculated by summing all principal components. On the other hand, the seismic source model, fragility curves, and consequence model given by Chaulagain et al. (2015) were used to evaluate the physical risk in OpenQuake. Similar to the study by Burton and Silva (2016), the integrated risk was evaluated using integrated risk modelling toolkit. Further discussion on social vulnerability, physical risk, and integrated risk assessment are presented in the following sections.

## 5.1 Discussion on social vulnerability assessment

In this paper, the objective of the social vulnerability assessment was to quantify the vulnerability in Nepal considering socio-economic parameters at the local level. The results of social vulnerability show that the most socially vulnerable places are located in the Far-Western, eastern Terai region, and western Terai region of Nepal. Our findings exhibit differences in social vulnerability in areas located in the same ecological region. The main reason behind this could be the pre-existing conditions like infrastructures, education, economy, etc. The population in the Far-Western region and the eastern Terai region are mostly minorities, Dalits, and marginalized groups who are behind in education and development (Gautam, 2017). As for Mountainous and Hilly areas in the Far-West region, the geographical terrain has affected the development path of these areas. Aksha et. al (2019) and Gautam (2017) also found a similar vulnerability in their respective studies. However, Aksha et. al (2019) classified Kathmandu valley as a high vulnerability class, while Gautam (2017) it as a very low class. Our social vulnerability result agreed with the latter case. This variability in the result is due to differences in variables and hazards considered during the analysis. Moreover, more recent data for SoVI will depict the more exact status of society and its vulnerability to disaster. This study uses Census data from 2011. And, the census is done every 10 years in Nepal, and the most recent census was held in 2021. However, data from 2011 were considered due to the unavailability of recent comprehensive data. More recent data can be used in future studies, once the Nepal census 2021 is published and made available.

## 5.2 Discussion on physical risk assessment

The objective of physical risk assessment in this study was to evaluate the physical risk index using a probabilistic approach. As in Burton and Silva (2016), the Classical PSHA risk-based calculator was used to assess loss exceedance curves, risk maps, and average annual loss. In this study, a probabilistic approach and region-specific steps were used to evaluate the seismic hazard curves as in Chaulagian, et al. (2015).

The probabilistic method of estimating seismic hazards used in the study utilizes the Poisson distribution model. Although earthquakes are assumed to occur randomly in space and time, the Poisson model assumes that earthquakes make a stochastically independent sequence of events in space and time (Anagnos and Kiremidjian, 1988). Despite such counterintuitive characteristic, the Poisson model is widely used due to its simplicity in the formulation and smaller range of parameters to be estimated. Moreover, the recent research (Weatherill et al., 2015; Schiappapietra and Douglas, 2020) in seismic risk assessment have incorporated spatially-correlated distribution not only to estimate simultaneous intensity measure levels at locations during a specific earthquake but also to quantify the correlation between locations. The present studies have suggested modelling of spatial correlation of earthquake ground motion since attenuation of ground motion is not only period-dependent but also regionally dependent. However, in our study, we have used the conventional method of probabilistic seismic risk assessment due to its simplicity. Nonetheless, a certain standard approach is necessary to evaluate comparable estimates of seismic hazards. Moreover, the authors are aware of the fact that numerous estimations such as casualties, non-structural damage, business interruption loss, and loss to critical infrastructure may improve the indicator of physical risk. However, only economic losses to buildings were utilized in this study as an initiation for this type of research for Nepal. Similarly, the results of physical risk (average annual loss estimates) were rescaled using the MIN-MAX method as mentioned in previous sections. The rescaling is necessary to integrate social vulnerability with physical risk although the rescaling of the estimates may have resulted the loss of spatial information of physical damage results.

## 5.3 Discussion on integrated risk assessment

The integration and mapping of the spatial distribution of average annual losses and social vulnerability is very useful. However, the integrated maps do not reflect the true effect of components inducing seismic risk at a particular location. This can be due to the compounding nature of the spatial risk as the areas of medium to high levels of social vulnerability

compound moderate levels of physical risk to generate high levels of integrated risk. The medium level of social vulnerability in the eastern Terai region is compounded with the high level of physical risk to create a higher level of integrated risk which can be seen in Figure 14. On the other hand, there is a higher degree of seismic risk and integrated risk in Kathmandu valley although the social vulnerability results depict lower degree of vulnerability. In light of the limitations of this study, it is clear that robust procedures and methods should be used in future analyses of integrated risk assessment. Although this study is accompanied by certain shortcomings, it is within the context that the inclusion of a higher number of factors that contribute to the mitigation of earthquake risk provides better approaches in the development of policy and plans to reduce overall seismic risk.

**6 Conclusion**

The impacts of earthquakes cannot be defined only from the potential damages from them. Such effect also depends on the capacity of society to address and rebound from damage. Social vulnerability index depicts how society will prepare and respond to any disaster, while seismic risk index (AAL) shows how society will get affected due to earthquakes. This paper presents an integrated study using SPSS and OpenQuake to delineate integrated seismic risk for Nepal. The integration of seismic risk with social characteristics gives a different outlook on seismic risk mitigation and planning. The major conclusions of this study are described below:

- The Far-Western, eastern Terai region, and western Terai region of Nepal were determined highly vulnerable from social vulnerability analysis. The main reason behind the differences in social vulnerability within same ecological region could be the pre-existing conditions like infrastructures, education, economy, etc. Moreover, the population residing in the Far-Western region and the eastern Terai region are mostly minorities, Dalits, and marginalized groups.

- Integrated risk helps clearing up the confusion on whether to focus on loss and damage, or the population that are least likely to be able to recover from losses. For example, if only social vulnerability index is considered, western Hills and Mountain region seems more vulnerable than Kathmandu Valley, while considering seismic risk index, Kathmandu Valley is more vulnerable. Only on integrating, we can confirm that Kathmandu Valley is more vulnerable to earthquakes and need more attention than western Hills and Mountain region.

- From the results of seismic risk assessment, Kathmandu valley and the eastern Terai region were determined as high seismic risk areas. Similarly, the integrated risk results indicated high vulnerability in Kathmandu valley and entire Terai region. The Far-Western Hills and Mountain regions were determined as the low vulnerable region as per integrated risk maps, even though they have a high SoVI index. These regions (Far-western Hills and Mountain) should still be depicted as major concern. This is because these areas might have a smaller number of houses but they can be of low quality, which could suffer damage to even low-magnitude earthquakes, and these regions are considered backward areas of Nepal in terms of infrastructure and development activities.

- The findings reinforce the concept in the hazards and vulnerability field that analysis of socio-demographic characteristics, when considered along with the physical environment, brings a greater understanding of the potential impacts of hazards.

- Additionally, this study provides a basis for local policymakers to integrate knowledge about the physical environment, social, and demographic composition of their region to assess their natural hazards mitigation, using a standardized tool like OpenQuake before an event occurs.

In this study, we assess social vulnerability characteristics and potential risks from a large earthquake on seismically active zones across the country. Since local level policymakers and municipalities have big responsibility to minimize, prepare, and respond to hazards and their impacts, a proper understanding of the social vulnerability is crucial to alleviate

the risks caused by earthquakes. The distribution of potential seismic hazard-related losses across the country can be partially explained by the region's ethnicity, income, and renter population. Although previous studies have also identified the integration or relationship between natural disasters and vulnerability features, this research extended the applicability of social vulnerability by integrating it with earthquake risk estimates.

## Author contribution

SB and RM initiated the research; SB gathered the data; SB and RM analysed the data; SB plotted the maps and graphs; RM wrote the manuscript draft; SB and RM reviewed and edited the manuscript.

## Competing interests

The authors declare that they have no conflict of interest.

## Acknowledgements

The authors would like to express special thanks of gratitude to Department of Civil Engineering, Pulchowk Campus for noteworthy support in this project. Also, the authors would like to thank Professor Henri P. Gavin from Duke University, Civil Engineering Department for mentoring during the process of seismic hazard and risk analysis.

## Data Availability

The dataset used in the study were derived from the following resources available in the public domain: National Population and Housing Census 2011 (CBS (2012)), Population Monologue V01 (CBS (2014b)), Population Monologue V03 (CBS (2014a)), Nepal Human Development Report (Sharma et al., 2014), Department of Health Services (2013), Budget report for year 2070-71 (2013 - 14), and Department of Education (2013 – 14).

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
