# Peer review of "Integrated Seismic Risk Assessment in Nepal"

_Natural Hazards and Earth System Sciences, 2021_

## Referee Comment (RC2)

The paper gathers a large amount of work and develops several ideas, it is thus dense and long. The paper content is relevant for publication in NHESS. However, due to the several comments and disorganized structure of the paper, major revisions are needed to proceed to future steps in publishing this work.

I have several comments that may however require some profound modifications in the manner the paper is structured. If the majority of these comments are not solved in the updated version of this paper, then it should be rejected for publication.

**A. General comments**

1. I recommend this paper goes through an extensive English edition and proofreading. There are several grammatical errors that must be fixed.
2. The paper claims in several parts that a method is presented. According to my understanding, that is not the case simply because there is no new method presented. The authors make use of local datasets along with already existing approaches and software. Thus, although they provide important results, they should be careful with the manner the word "method" is used. As a major comment, the authors should spend more elaborated comments in an actual discussion.
3. Aligned with the former, I strongly recommend you to present a new big section (same level as the introduction, results, or conclusions) named "discussion" (different from the merged one "results and discussion", which should be split). In this chapter you should present a detailed comparison of your results with existing works, as you have nicely done with Aksha et al, 2019 and moderately (not really clear) with Gautam,2017. Still, other studies are still missing (e.g. the hazard and risk model of the Global Earthquake Model (GEM) from Silva et al, 2020). Therein the authors are expected to write the main drawbacks of their assumptions, how limitations could be overcome in future studies, and what would be needed to achieve more accurate results.
4. The conclusions should be more elaborated and precise. Please consider including bullet points highlighting your main findings.

**B. Specific comments**

Please note that in the following comments, some references are suggested to be included whilst, others, due to their irrelevance to the overall aim of the paper, are suggested to be removed

1. "Abstract"
   1.1. The Abstract should be considerably reformulated. Starting with the particular study case and then the general problem is not a recommended manner to write it.
   1.2. Please avoid using acronyms in the Abstract (VDC). Also, that is never defined. Please do it for the first time it is presented anywhere in the text (again, not in the abstract).
   1.3. Line 8: "annual loss (seismic risk)" is not an accurate manner to describe the final metric of interest. Please rewrite it.
   1.4. Line 9: the expression "the earthquake risk" should be changed to something concrete saying what are the exposed assets of interest in your study.
   1.5. Line 10: the expression "compute the risk analysis" is not correct. Better saying "assess the risk to seismic ground shaking of the exposed residential buildings in Nepal". This narrows down the scope of your paper since you are not considering any other type of exposed assets

or other compound and secondary hazardous effects (e.g. ground failure, liquefaction, co-seismic landslides).

**1.6.** Line 11: Secondary aspects such as the Software (Openquake and ArcGis 10) that you select for your objective are unnecessary details at this very general and early stage of the paper. These details should only be provided later on once you have elaborated more on your ideas.

**2.** "1. Introduction"

**2.1.** Line 30: you provide a general detail at the global scale right in between two sentences concerning only Nepal. This is distracting. Please consider deleting or relocating it. The same happens with the content of lines 32-35. You could fix the former issues as follows: since you invest quite a long text in then describing general aspects of the risk assessment practices, and only the local study area is retaken in line 50, I suggest separating the global from the local aspects. In this case, the first part of the introduction should be relocated. Please use a different paragraph when you introduce such a change of setting.

**2.2.** The ideas that are introduced in line 50 are not clear. Why and how do you justify that the seismic hazard mapping in Nepal is scarce? (You use "lack"). There have been several recent projects done in the same study area, some of them are not old, and you cited them. Thus, please rephrase it.

**2.3.** Line 59: "risk assessment with proper forecasting measures" is not clear. Perhaps adding some example measures might help to provide clarity on what you exactly are trying to refer to. If these "measures" are the explanation you provide in lines 60-61, then I suggest you deeply reformulate that sentence.

**2.4.** Line 62: The OpenQuake reference is cited only in line 97, but disregarding its inadequate use in the Abstract (as formerly described), it was also mentioned in line 62. A proper reference is needed there. Subsequently, in line 63 it is only mentioned that the formal analysis you carry out is *"in the districts of Nepal".* The reality is that you do a much more exhaustive analysis of the municipalities. I suggest rewriting it.

**2.5.** Since some readers might not be familiar with the Nepali administrative division, I suggest you explicitly say right after the first time the word "district" is mentioned what does it exactly mean (i.e. which level of division is it and what is its average area (considering the 75 ones)?)

**2.6.** The sentence comprised between lines 64 and line 65 "*Studying social vulnerability identifies the sensitive areas and populations that are prone to high risk and are less likely to acclimatize and recover from a natural catastrophe"* is disconnected from the previous paragraph. This is background information that should be provided somewhere else. Also, I do believe the terminology *"natural catastrophe"* is incorrect. Please modify carefully using the correct terms. The same comment applies to "seismic losses" (used twice in the text). Please replace it with something more concrete.

**3.** "2. Theory and background"

**3.1.** Please read alone the sentence: "The impact of natural hazards is based on social parameters such as socioeconomic status, geographical features, ethnicity (minority), renter, gender, and age". The manner it is written could be misunderstood in the sense that the listed variables are some of the most important ones, disregarding the hazard component. That should be also clearly said, e.g. beside the hazard component, there are others…). Moreover, that

sentence needs a reference. You could simply consider correctly rewriting this sentence and the former one and use the citation therein provided.

**3.2.** The sentence "There have always been stories of high-class predation and low-class vulnerability" and the next one "At the same time… society" are prone to open free interpretations. Although I agree with what the authors try to express, since these sentences lack citation and explicit examples, due to the manner they are written, they remain as opinions whose locations are out of context in a "theory and background" section. Please remove them, or better, rewrite them accordingly.

**3.3.** The text in lines 85 (related to population) and 86 (related to topography) are presented as mere ideas in a disconnected manner from the previous (and nicely written text). Please use connectors or create a smoother manner to connect those ideas with each other.

**3.4.** Please remove "OpenQuake" from the header 2.2. This is totally irrelevant for a title. Furthermore, I strongly suggest removing the short description of that software from this title. The user's decision about the desired tool to implement is not part of "theory and background", but rather on the "material and method" section.

**3.5.** The authors should be careful in the manner the first sentence of Section 2.2 is presented. If you note the cited work of Stevens et al., (2018), they say: *"a **large part** of Nepal **is expected** to have **a 10% probability of exceeding shaking** of 0.4g–0.6g and a 2% probability of exceeding shaking of 1.0g–3.0g in any 50 year period".* Please note that it refers that only one portion may experience such ground motions, not all the country. This is totally missing in your sentence. Also, please note that in this context the conditional form is advised (e.g. is expected), whilst you use "is". That is incorrect and unacceptable. Therefore, I also suggest to rephrase these sentences in a more pragmatic manner (e.g. accelerations in the range of XXX may be expected for a 10 % probability of exceedance over 50 years or 475 years return period for certain zones of the study area). The same could be rewritten for the other probability of exceedance. Furthermore, as you certainly know, this type of formulations assume a Poissonian mode, and hence, you might like to highlight that important (and nowadays questionable) assumption herein.

**3.6.** The authors should state in both sections (3.2.2. and the to-be created Discussion) the implications of not having used any spatial correlation model to model the seismic ground motion fields. Justify your assumption being aware of this limitation, for this aim, please consider citing Weatherill et al, (2015)

4.  **"3. Materials and methods"**

**4.1.** Line 100: *"potential effects"* is too vague. Please be (way) more precise. You say "This approach", but which approach? You have not even mentioned the method you will follow. You simply mentioned "an integrated approach" and then cite Burton and Silva, 2016. I strongly suggest that you reformulate the paragraph explicitly indicating that you will use an existing method (e.g. Burton and Silva, 2016) to integrate physical vulnerability to seismic ground shaking with other "human" dimensions. Please remove "within the hazard zone" This is simply not clear nor accurate. The sentences you use to cite Carreño, et al, 2012 and Fernandez et al., 2006 are both presented as background information. That is incorrect for the type of title you are describing. Thus, please consider to either moving it to the Introduction, or to rephrase it saying that you will use a method presented by those authors.

**4.2.** "3.1. Social vulnerability assessment"

    4.2.1. The Section "3.1.1 Data and SoVI modification" should be renamed. It is just too vague. Is this input data? For what procedure? Please indicate it in the title. Avoid the use of acronyms here.

    4.2.2. Line 120: you say 54. Is this a mistake and do you refer to 45?

    4.2.3. Line 116: For the sentence *"Table 2 provides the list of all the variables used for social vulnerability assessment",* please specify whether those variables are a result of your subjective selection among more, or are they all integrally (not removing any) within each dataset employed. This is not clear.

    4.2.4. Line 131. The subscript in "Si" is incorrect. Please change it.

    4.2.5. Figure 1:

        4.2.5.1. The location within the text of this figure is not accurate. Please note it belongs to the section "Data and SoVI modification" within "Materials and method", which is weird. This figure does not really display any important variable of analysis for these sections, but only show the geographical location of some three regions and administrative divisions. Thus I suggest this is included within a more generic title "Context of the study area" or similar. This is not a must, but it is highly advised. In this case, others parts should be relocated therein.

        4.2.5.2. If possible, please remove from the legend "_". This is not harmonious.

        4.2.5.3. Please add the borders with neighboring countries. The way it is shown it shows Nepal as an island.

        4.2.5.4. Please reduce the font size of the coordinates. Having them larger than the actual caption is not aesthetic.

        4.2.5.5. For guidance, please consider adding the location of some of the main cities.

    4.2.6. Section "3.1.2 Principal component analysis" should be renamed adding for which sub-process this method is planned to be done, otherwise remains unclear.

    4.2.7. Line 163: The manner the authors describe old versions of the method is irrelevant to the main Section "Materials and methods". Also, right next to it and to remark a difference with newer approaches, the word "currently" is used while citing outdated references (1993-2000). If that is anyway your intention, please provide current (from this decade) references.

    4.2.8. Line 185: "test value of 0.000". Really? 0? Please check it out. Although it is true that the number of decimals in all of the provided numbers should remain equal (please work on this), the cero can be an exception (Although I suspect it is unlikely it is a mere null value).

    4.2.9. Line 186: Please provide the reference and full name (Statistical Package for the Social Sciences of SPSS v.21.0.

    4.2.10. Line 191: incorrect manner of presenting the citations. Open a parenthesis after "studies" that contains both references.

    4.2.11. Once again, the manner some comments provided as background are provided (e.g. SoVI scores are generally expressed as standard deviations (z-scores) or quintiles to emphasize their relative value (Tate, 2012)") within the formal analysis and explained steps is highly distracting. Please rephrase this sentence and all of the ones with similar style. You can say: "Similarly to XXX, we expressed the SoVI scores as YYY…."

**4.3.** "3.2. Seismic Risk Assessment"

4.3.1. In this paragraph you are mixing up present and past tense without any order, please make sure that during the English proofreading this is harmonized.

4.3.2. Line 202. Please rewrite or delete this sentence "The physical vulnerability function is solely hazard dependent". If you decide to rewrite it, please add a citation, but not in using a "background style" as formerly discussed.

4.3.3. Information contained in lines 207 until the first point of 209 is irrelevant for the subsection. Also, it is badly written and there is no order with the (unnecessary) provided references. They must be ordered following certain selected pattern.

4.3.4. For the section "3.3.1 Source model", please rewrite the information about the former studies of seismic source zonations making a link with the actual existing model you decided to adopt. The way it is presented can be misleading due to the expression "for instance". You can extensively simply this saying something like: Despite there have been several seismic sources for the study area (i.e. ZZZ, UUU, and TTT), we have decided to implement the one of XXX that comprises 23 area source" or something similar. Also, in this section, you should mention the main tectonic setting of the region, the Main Himalayan Thrust, which you neglect in your text.

4.3.5. Line 218: please change the expression "effect on engineering structures". This is not accurate. I do understand what you try to say, but, please be more precise in a written form.

4.3.6. Line 219: specify the type of Earthquake magnitude you are referring to. I suspect it is moment magnitude (Mw), but it can be confusing due to the two types of magnitudes provided in Table 5. Thus, this aspect must be clearly stated therein. The expression "the surface wave of 4.0" is inaccurate, please rewrite it. Be careful with the manner you write such technical details.

4.3.7. Line 220: the list of devastating earthquakes is not necessary here. The information of Table 5 is sufficient. Also, this is repeated information (already provided in line 26 (with a proper reference).

4.3.8. If possible, please modify Figure 3, reducing the fontsize of the coordinates, including neighboring countries, and most importantly, since the authors have decided to not include a single map showing the seismic hazard, at least, this figure containing the zonation should contain some of the most important historical earthquakes (similarly as done Thapa and Guaxin, 2014 and Chaulagain et al, 2015). This way, the reader can realise that despite there are more assets exposed in the southern parts of the country, the historical seismicity is comparatively lower. This type of comment is also missing. Please consider including something similar.

4.3.9. Related to the former, please consider citing the study of Rao et al, 2020 where the extend of likely synthetic ruptures are presented for Nepal (see figure 1 in their study)

4.3.10. The first column of Table 5 is unnecessary because the zones have the same numeration. Please delete it. Also, since this information is exactly the same as the Chaulagain et al, 2015 which in turn took it from Thapa and Guoxin 2013, if you really want to keep this table, you should use a word highlighting that there is no contribution from your side in their elaboration. Such a word can be "taken or reprinted", but I feel that a simple citation is in this case not necessary.

4.3.11. Line 231: The plural form of research will also be "research". Please change it.

4.3.12. Line 238: I dislike the way the building types are included in the exposure model. Considering that according to the GEM V.2.0 taxonomy, there are other several occupancy types for buildings different from the industrial and commercial (e.g. education, government, assembly, agriculture), that would mean that all of those typologies are implicitly included in your model, which I doubt. If only residential buildings were included, I suggest you explicitly say it. Otherwise, it is not clear.

4.3.13. Line 240: *"most of the regions in Nepal consist of"*. Please change the word "regions" and be more specific (replacing it to "the residential building stock" or similar).

4.3.14. Since the authors are not doing any difference in the loss to damage ratios of the consequence model for each building type, the information contained in Table 7 is not interesting enough to be shown as a table. I suggest to reconfigure such data as simple text.

4.3.15. Line 255: There is an imprecision in the sentence "The fragility function used in this study is shown in Table 6". What the authors report are the logarithmic mean ($\mu$) and logarithmic standard deviation ($\sigma$), not the "fragility functions". Moreover, the definition of the nomenclature $\mu$ and $\sigma$ is missing. Please include it either in the text or table caption. Moreover, I could not find anywhere where it is explicitly stated the intensity measure(s) that those fragility functions employ. Are they all "working" with PGA, PGV, any other spectral acceleration, or more sophisticated ones? Please clarify.

4.3.16. I highly recommend to include a figure showing the graphical representation of these fragility functions. This will provide clarity because, due to the logarithm values, it is not straightforward for the reader to make out an idea about the differences between the damage states across the 5 building types. Use gridded subplots for this aim.

4.3.17. Figure 4: The subplot (a) must be reformatted. Please avoid using "_" in the legend. The used ranges to define the building counts are inaccurate. Please note how the numbers are not consecutive. This is simply unacceptable. A good practice can be found in one study you have cited: please refer to Figure 3 in Chaulagain et al, 2015 to realise how the numerical ranges must be consecutive. Please avoid using white as a color as well as colors with sharper distinctions. Instead, I strongly suggest to use a graduated color map (similar to Chaulagain et al, 2015). Make sure that the colors employed are "color-blind" friendly.

**4.4.** "3.3 Integrated risk assessment"

4.4.1. Please introduce the text and then, figure 5. It is weird to have it before their actual formal introduction.

4.4.2. Also, regarding figure 5, I strongly suggest to modify it. It is not self-explanatory. I am aware you have based your figure on the existing Fig. 1 of cited Burton and Silva, (2016), but you should use an expression saying you that you fully relied on theirs to produce yours (i.e. "modified from" or similar). Please note that theirs is indeed self-explanatory because they added a short explanation for the components A, B, C, and D. Please consider doing something similar. Some hints are:

4.4.2.1. For the seismic hazard, it is important to remark it is probabilistic hazard and not scenario-based.

4.4.2.2. For what you call "geographic features" it is important to remark what exactly you refer to. In this sense, although Burton and Silva, (2016) mentioned Exposure and physical vulnerability, you should mention it, but even narrowing down saying "Exposure modelling for residential buildings and their physical vulnerability".

4.4.2.3. Complementary, a senctence where you raised awareness that your "exposure products" are part of modelling process with uncertainties and not a ground truth. This type of sentence should be include. Please consider including these references: Kalakonas et al, (2020) and Gomez-Zapata et al, (2022) for this aim.

4.4.2.4. Moreover, a statement about the geographical part of the exposure model is missing. Although it is implicitly said to be modeled using administrative boundaries, the implication of using any other type of aggregation areas for risk assessment could have been another alternative. Studies such as: Douglas, 2007, Bal et al, 2010; and Gomez-Zapata et al, 2021 have shown the importance of having compatible with the hazard footprint and attenuation. You could briefly present these issues and references either here or in the discussion section.

4.4.2.5. For the "physical risk" you should clearly show what is/are the metrics (similarly as the aforementioned authors did).

4.4.3. The first three lines of this section are presented as background within a "material and methods" section. This structure is inaccurate and instead, it should be relocated somewhere else (e.g. Introduction). Also, please be aware this is repeated information (see where else you have cited Burton and Silva, (2016)) that the reader has already come across several times. I agree that this citation is highly relevant, but you could anyway consider to reduce the number of times that similar ideas that come out of this reference are mentioned in your paper.

**4.5.** The sentence *"Here, the seismic losses were recomputed by using the Min-Max rescaling method"* is out of context within the current title. That is something the authors had already correctly presented in Section 3.2 (line 205). If your intention is to mention that the outcomes of the former step are integrated in this one, then you must properly rewrite that sentence.

**4.6.** Similar issue as described above regarding the sentence "The seismic hazard analysis requires earthquake ruptures and ground motion fields". It is unnecessary and also out of place. This is something you have already elaborated in sections 3.3.1. 3.2.2. If you still want to keep it, you must properly rewrite it.

**5.** "Results and discussion"

**5.1.** "As clearly stated in the beginning of this revision, please separate "results" and "discussion" into two differentiable sections.

**5.2.** "4.2. Seismic Risk Assessment".

5.2.1. The authors mentioned since section 3.2 that Chaulagain et al, 2015 also performed a seismic risk assessment for the same study area. Only a short comment on a single similarity is mentioned. However, detailed differences with that study (or others) are not provided. This is key in the reconnaissance of the added value of your results (even without mentioning the nice contribution about social vulnerability). This should be included in the new section "discussion" I am kindly asking for.

5.2.2. Line 308: Although some names of the regions were formerly provided in Fig. 1, this one should be recalled one again. It is easy for the reader forget it. However, some names you cite have never been introduced (i.e. Kathmandu, Butwal). This is one of the reasons of the comment 4.2.5.5 related to Figure 1. Same comment applies for line 238.

5.2.3. Figure 11: The caption is incorrect. It is true that you present the results only highlighting the 75 district boundaries, the results that you are displaying are in reality the ones for the 3983 VDCs and municipalities. Therefore, I strongly recommend that you rewrite the caption. Do not be afraid of including more than 1 line descriptions when they are necessary to be fully understandable to the readers.

**6.** Conclusions

**6.1.** Line 351: *"This paper presents a method using…"* (This is related to one of the general comments described above). This is because no original method was outlined. Instead, the authors made use of existing approaches to present some results. Please rewrite it. Be precise.

**6.2.** The last paragraph of the Conclusion should be moved to this new "discussion" section. Complementary, considering that some preliminary reports of the 2021 Nepal Census are already available by April 2022, the authors are encouraged to discuss (in the future "Discussion" section) how a similar analysis as they performed using outdated information from 2011 could locally or globally vary if such a new census was used instead.

**C. Technical corrections**

1. Most of the references lack the correct style that is demanded by the Journal. The DOI of the references must be provided in the next revision.
2. Line 54: incorrect use of the parenthesis for the citation "Aksha et al., 2019"
3. Table 1 provides sensitive information but without providing any reference. Please add it to the caption.
4. There are several identical titles (e.g. "3.2 and 4.2"; 3.3 and 4.3). A distinction must be made at least either mentioning the study area or something else (method vs. results).
5. Line 289: please replace the word "total" by "integrated". This is to be consistent with Fig. 5
6. Line 293: please remove or rewrite "The OpenQuake platform present within QGIS was used to develop probabilistic seismic risk models". This suggestion comes from the fact that there is a very similar sentence just before this one. Please avoid redundancy.
7. Figure 6: Please remove "_" from Figure 6. Please ensure that the notation for the building types is identical to the one provided in Table 6. Also, please modify the caption. Saying "QGIS IRMT" is not accurate. IRMT is enough. Since captions should be self-explanatory, please rewrite the full name of this acronym and include its respective citation.
8. Line 304: Please modify "geophysical characteristics". Due to the various use of that word in other related scientific contexts, you should use other wording.
9. Line 366: "SIn": This is wrong. Please correct it.

**D.  Full references that are suggested to be included in the updated manuscript after this review:**

- Bal, I. E., Bommer, J. J., Stafford, P. J., Crowley, H., and Pinho, R.: The Influence of Geographical Resolution of Urban Exposure Data in an Earthquake Loss Model for Istanbul, Earthquake Spectra, 26, 619–634, https://doi.org/10.1193/1.3459127, 2010.

- Chaulagain, H., Silva, V., Rodrigues, H., Spacone, E., and Varum, H.: Earthquake loss estimation for the Kathmandu valley, Second European Conference on Earthquake Engineering and Seismology (2ECEES), 2014

- Contreras, D., Chamorro, A., and Wilkinson, S.: Review article: The spatial dimension in the assessment of urban socio-economic vulnerability related to geohazards, Natural Hazards and Earth System Sciences, 20, 1663–1687, https://doi.org/10.5194/nhess-20-1663-2020, 2020

- Douglas, J., 2007. Physical vulnerability modelling in natural hazard risk assessment. Natural Hazards and Earth System Sciences 7, 283–288. https://doi.org/10.5194/nhess-7-283-2007

- Gomez-Zapata, J. C., Brinckmann, N., Harig, S., Zafrir, R., Pittore, M., Cotton, F., and Babeyko, A.: Variable-resolution building exposure modelling for earthquake and tsunami scenario-based risk assessment. An application case in Lima, Peru, 21, 3599–3628, https://doi.org/10.5194/nhess-21-3599-2021, 2021.

- Gomez-Zapata, J. C., Pittore, M., Cotton, F., Lilienkamp, H., Simantini, S., Aguirre, P., and Hernan, S. M.: Epistemic uncertainty of probabilistic building exposure compositions in scenario-based earthquake loss models, https://doi.org/10.1007/s10518-021-01312-9, 2022.

- Kalakonas, P., Silva, V., Mouyiannou, A., and Rao, A.: Exploring the impact of epistemic uncertainty on a regional probabilistic seismic risk assessment model, Natural Hazards, https://doi.org/10.1007/s11069-020-04201-7, 2020.

- Rao, A., Dutta, D., Kalita, P., Ackerley, N., Silva, V., Raghunandan, M., Ghosh, J., Ghosh, S., Brzev, S., and Dasgupta, K.: Probabilistic seismic risk assessment of India, 36, 345–371, https://doi.org/10.1177/8755293020957374, 2020.

7.  Silva, V., Amo-Oduro, D., Calderon, A., Costa, C., Dabbeek, J., Despotaki, V., Martins, L., Pagani, M., Rao, A., Simionato, M., Viganò, D., Yepes-Estrada, C., Acevedo, A., Crowley, H., Horspool, N., Jaiswal, K., Journeay, M., and Pittore, M.: Development of a global seismic risk model, Earthquake Spectra, 8755293019899953, https://doi.org/10.1177/8755293019899953, 2020.

8.   Weatherill, G. A., Silva, V., Crowley, H., and Bazzurro, P.: Exploring the impact of spatial correlations and uncertainties for portfolio analysis in probabilistic seismic loss estimation, Bulletin of Earthquake Engineering, 13, 957–981, https://doi.org/10.1007/s10518-015-9730-5, 2015.

---

## Author Comment (AC2)

Thank you very much for your extensive review of our paper including comments and suggestions. A major revision of the paper has been carried out to take account of all the comments. We believe, during the process, the paper has been significantly improved.

We have sequentially addressed all of the points raised by the referee in each section of the paper along with the general and technical comments. The major changes include reformulation of abstract, inclusion of discussion, and correction of the presentation of maps. The revised maps are added in the last section of this document.

Major changes:

**1. Abstract**

Seismic risk assessment involves combination of the exposure model, seismic hazard, and structural vulnerability. The 
[revised manuscript text omitted]
 from spatial information loss of physical damage results. This study presents the integrated risk due to earthquakes in sub-divisions administrative units of Nepal i.e., districts and VDCs. The mapping of the spatial distribution of average annual losses and social vulnerability is very useful, but it doesn't reflect the true effect of components inducing seismic risk at a particular location. This can be due to the compounding nature of the spatial risk as to the areas of medium to high levels of social vulnerability compound moderate levels of physical risk to generate high levels

of integrated risk. There is a higher degree of seismic risk and integrated risk in Kathmandu valley whereas, the medium level of social vulnerability in the eastern Terai region is compounded with the high level of physical risk to create a higher level of integrated risk which can be seen in Figure 11. In light of the limitations of this study, it is clear that robust procedures and methods should be used in future analyses of integrated risk assessment. Although this study is accompanied by certain shortcomings, it is within the context that the inclusion of a higher number of factors that contribute to the mitigation of earthquake risk provides better approaches in the development of policy and plans to reduce overall seismic risk.

**3. Title changes:**

2.2 Earthquake risk assessment to Parameters of Earthquake Risk Assessment

3.1.1 Data and SoVI Modification is changed to Social Vulnerability Indicators.

3.1.2 Principal component analysis changed to Calculation of SoVI (Social Vulnerability Index) by Principal Component Analysis (PCA)

3.2 Seismic Risk Assessment changed to Evaluation of Physical Risk Assessment

4.1 Social vulnerability index (SoVI) changed to Results of Social Vulnerability Analysis

4.2 Seismic Risk Assessment changed to Results of Seismic Risk Assessment

4.3 Integrated Risk Assessment to Results of Integrated risk assessment

**General Comments**

→ We agree that there are grammatical errors within the text, we have tried to correct them to the fullest. The words "methods" and "method" that were mentioned in the paper have been substituted. We have added a big section called "Discussion" prior to conclusion. The detailed comparison of the study with past studies has been carried out. In addition, we have also included limitations and steps to overcome these setbacks.

**Abstract**

→The abstract has been reformulated. All of the points are addressed in the reformulated version of abstract.

**Introduction**

→The introduction has been organized as per suggestions mentioned above. Comment 2.1: The national and global aspects of earthquakes have been separated. Comment 2.2: The text "lack of seismic mapping" has been replaced with "lack of social data for analysis and mapping." Comment 2.3 The sentence has been paraphrased. Comment 2.4 The OpenQuake reference has been cited. Comment 2.5 The administrative division of Nepal has been introduced. Comment 2.6 As mentioned, we have rewritten the section according to the comments.

**Theory and Background**

→**Comment 3.1**: Reformulated sentence as per suggestion: "Besides the hazard component, there are other social parameters that determine the impact of natural hazards such as socioeconomic status, geographical features, ethnicity (minority), renter, gender, and age (Burton and Silva, 2016)." **Comment 3.2**: We have removed the subjective statement "There have always been stories of…" **Comment 3.3 and 3.4**: Reformulated as per suggestion. Comment 3.5: Included in discussion part.

**Materials and methods**

→**Comment 4.1**: Reformulated the first sentence to: "This study assesses seismic risk by combining it with the human dimensions within the hazard zone similar to that in Burton and Silva (2016). This approach is an integrated seismic risk assessment." **Comment 4.2.2 & 4.2.3**: Table 2 provides the list of all the variables used for social vulnerability assessment. Out of 45 variables, district-wise indicators were represented by 22 variables and each sub-section (municipality and VDCs) was assumed to have uniform value. Among these 45 variables, seven of them were weighted combination of multiple variables as shown in Table 2. These weighted variables were obtained from 54 variables mentioned in Table 3. Therefore, altogether 92 variables (45-7+54) were considered for SoVI index. **Comment 4.2.5**: Figure 1 has been moved to the introduction as a part of background information and study area description and the background information in this section has been moved to the theory part.

**Comment 4.2.8**: Yes, we have rechecked and we obtained the test value of 0.000, as shown below. Similarly, we found some other studies with same cases. For example, the study by Yogamalar and Samuel (2018)[1].

**KMO and Bartlett's Test**

| Kaiser-Meyer-Olkin Measure of Sampling Adequacy. | | .888 |
|---|---|---|
| Bartlett's Test of Sphericity | Approx. Chi-Square | 258644.625 |
| | df | 1035 |
| | Sig. | .000 |

**Comment 4.2.11:** Reformulated statement: "As presented in the paper Tate (2012), SoVI scores were used in the form of standard deviations (z-scores) or quintiles to emphasize their relative value." All other comments of section 4.2 and 4.3 have been addressed in accordance to the suggestions. Some important changes are: **Comment 4.3.12:** Reformulated statement: "In this study, the building description and data from Census 2011 were used to develop the exposure model without considering the industrial or commercial buildings. In other words, only residential buildings were considered for the exposure model. The exposure models used in the study are part of seismic risk assessment with uncertainties, although present studies like Kalakonas et al. (2020) and Gomez Zapata et al. (2022) have pointed how the epistemic uncertainties embedded in exposure models are propagated throughout the computation of seismic risk. We have considered five types of residential buildings —……" It is also mentioned in "Discussion" section that "The authors are aware of the fact that numerous estimations such as casualties, non-structural damage, business interruption loss, and loss to critical infrastructure may improve the indicator of physical risk. However, only economic losses to buildings were utilized in this study as an initiation for this type of research for Nepal." **Comment 4.3.15:** Fragility Function has also been added:
* * *
[1] Yogamalar, I., & Samuel, A. A. (2018). Classification of shared values for educational research. *International Journal of Human Resource Studies*, *8*(2), 47. DOI: 10.5296/ijhrs.v8i2.12818

[Figure]

**Figure: Fragility curves for a) adobe b) mud mortar c) cement bonded d) wooden buildings**

[Figure]

**Figure: Fragility curves for RCC buildings**

[Figure]

**Figure: Vulnerability curves a) adobe, mud mortar, cement bonded, and wooden buildings b) RCC buildings**

**Comment 4.4**: The section Integrated Risk Assessment has been reformulated as: "A total risk index was constructed by combining the social vulnerability index and estimates of average annual loss in rescaled metrics. The framework or workflow of the integrated risk assessment is shown in Figure 5. The first step in Figure 5, seismic hazard, was evaluated using the Probabilistic approach. The geographic features represent exposure modelling for residential buildings and their physical vulnerability. Combining these parameters, seismic risks were evaluated in terms of Average Annual Loss (AAL) which was further recomputed by using the Min-Max rescaling method. The physical vulnerability and exposure model interacts with the social and economic parameters or overall capacity of the population to sustain hazards (Burton and Silva, 2016). The social features define socio-economic parameters related to the demographic population to prepare for, react to, and recuperate from damaging events (Burton and Silva, 2016). The integrated risk is the combination of physical risk and a set of contextual and social vulnerability conditions (Carreño et al., 2012). In this regard, the paper evaluates the integrated risk grounded on direct factors or physical risk and socio-economic factors."

**Results, Discussions, and Conclusions**

We have reorganized results sections with different titles as mentioned above. A new section "Discussion" has also been added. The figures and results without further elaboration have been included in "Results" section whereas, the explanation, limitations, and suggestions have been included in "Discussion" section. Similarly, we have also modified "Conclusions" section as per the comments.

**Figures:**

Addressing Comment 4.2.5 (Figure1), 4.3.8 (Figure 3), 4.3.17 (Figure 4), C_7 (Figure 6), following changes are made.

➢ "_" are removed on legends.

➢ Borders of India and China are added.

➢ The font size of coordinates is reduced.

➢ Historical Earthquake data are added in figure 4.

➢ Numbers are made consecutive in figure 4.

➢ Captions are changed as per suggestions.

[Figure]

**Figure 1: Administrative Map of Nepal showing 3983 VDCs and municipalities, 75 districts, seven provinces with their headquarters, and three geographical regions**

[Figure]

**Figure 2: Seismic Source Zones along with spatial distribution of earthquakes in Nepal (Thapa and Guoxin, 2013).**

[revised manuscript text omitted]

---

## Author Response (AR1)

Thank you very much for your extensive review of our paper including comments and suggestions. A major revision of the paper has been carried out to take account of all the comments. We believe, during the process, the paper has been significantly improved.

We have sequentially addressed all of the points raised by the referee in each section of the paper along with the general and technical comments. The major changes include reformulation of abstract, inclusion of discussion, and correction of the presentation of maps. The revised maps are added in the last section of this document.

**Major changes:**

**1. Discussion**

The findings of this study can be briefly summarized into three parts: evaluation of the social vulnerability index, physical risk assessment, and integration of the social vulnerability & physical risk. The objective of the social vulnerability assessment is to quantify the vulnerability in Nepal considering socio-economic parameters at the local level. Based on z-scores, the total SoVI scores were classified into five quintiles from very low (< -1.5 standard deviation) to very high (> 1.5 standard deviations) vulnerability. The total SoVI scores were calculated by summing all principal components. The results of social vulnerability show that the most socially vulnerable places are located in the Far-Western, eastern Terai region, and western Terai region of Nepal. Our findings exhibit differences in social vulnerability in areas located in the same ecological region. The main reason behind this could be the pre-existing conditions like infrastructures, education, economy, etc. The population in the Far-Western region and the eastern Terai region are mostly minorities, Dalits, and marginalized groups who are behind in education and development (Gautam, 2017). As for Mountainous and Hilly areas in the Far-West region, the geographical terrain has affected the development path of these areas. Aksha et. al (2019) and Gautam (2017) also found a similar vulnerability in their respective studies. However, Aksha et. al (2019) classified Kathmandu valley as a high vulnerability class, while Gautam (2017) it as a very low class. Our social vulnerability result agreed with the latter case. This variability in the result is due to differences in variables and hazards considered during the analysis. Moreover, more recent data for SoVI will depict the more exact status of society and its vulnerability to disaster. This study uses Census data from 2011. And, the census is done every 10 years in Nepal, and the most recent census was held in 2021. However, data from 2011 were considered due to the unavailability of recent comprehensive data. More recent data can be used in future studies, once the Nepal census 2021 is published and made available.

Likewise, the objective of physical risk assessment is to evaluate the physical risk index using a probabilistic approach. As in Burton and Silva (2016), the Classical PSHA risk-based calculator was used to assess loss exceedance curves, risk maps, and average annual loss. There are several studies that show a varying range of seismic hazard analyses of Nepal. According to Stevens et al. (2018), in the large part of Nepal, the accelerations in the range of 0.4g-0.6g and 1.0g-3.0g may be expected for 10% and 2% probability of exceedance over 50 years period respectively. Chaulagain (2015) evaluated the estimated peak ground accelerations (PGA in g) at 10% and 2% probability of exceedance in 50 years in the range of

0.22-0.5 and 0.42-0.85g, respectively. Thapa and Guoxin (2013) estimated the PGA (in g) at 10% and 2% probability of exceedance in 50 years in the range of 0.21–0.62 g and 0.38–1.1 g, respectively. In this study, a probabilistic approach and region-specific steps were used to evaluate the seismic hazard curves as in Chaulagian, et al. (2015).

The probabilistic method of estimating seismic hazards used in the study utilizes the Poisson distribution model. Although earthquakes are assumed to occur randomly in space and time, the Poisson model assumes that earthquakes make a stochastically independent sequence of events in space and time (Anagnos and Kiremidjian, 1988) Despite such counterintuitive characteristic, the Poisson model is widely used due to its simplicity in the formulation and smaller range of parameters to be estimated. Moreover, the recent research (Weatherill et al., 2015; Schiappapietra and Douglas, 2020) in seismic risk assessment have incorporated spatially-correlated distribution not only to estimate simultaneous intensity measure levels at locations during a specific earthquake but also to quantify the correlation between locations. The present studies have suggested modelling of spatial correlation of earthquake ground motion since attenuation of ground motion is not only period-dependent but also regionally dependent. However, in our study, we have used the conventional method of probabilistic seismic risk assessment due to its simplicity. Nonetheless, a certain standard approach is necessary to evaluate comparable estimates of seismic hazards.

Moreover, the authors are aware of the fact that numerous estimations such as casualties, non-structural damage, business interruption loss, and loss to critical infrastructure may improve the indicator of physical risk. However, only economic losses to buildings were utilized in this study as an initiation for this type of research for Nepal. The results of physical risk and average annual loss estimates were rescaled using the MIN-MAX method as mentioned in previous sections. The rescaling is necessary to integrate social vulnerability with physical risk although the rescaling of the estimates may have resulted the loss of spatial information of physical damage results. The mapping of the spatial distribution of average annual losses and social vulnerability is very useful, but it doesn't reflect the true effect of components inducing seismic risk at a particular location. This can be due to the compounding nature of the spatial risk as the areas of medium to high levels of social vulnerability compound moderate levels of physical risk to generate high levels of integrated risk. The medium level of social vulnerability in the eastern Terai region is compounded with the high level of physical risk to create a higher level of integrated risk which can be seen in Figure 14. On the other hand, there is a higher degree of seismic risk and integrated risk in Kathmandu valley although the social vulnerability results depict lower degree of vulnerability. In light of the limitations of this study, it is clear that robust procedures and methods should be used in future analyses of integrated risk assessment. Although this study is accompanied by certain shortcomings, it is within the context that the inclusion of a higher number of factors that contribute to the mitigation of earthquake risk provides better approaches in the development of policy and plans to reduce overall seismic risk.

**2. Title changes:**

2.2 Earthquake risk assessment to Parameters of earthquake risk assessment

3.1.1 Data and SoVI Modification is changed to Indicators of social vulnerability assessment

3.1.2 Principal component analysis changed to Calculation of Social Vulnerability Index (SoVI) by Principal Component Analysis (PCA)

3.2 Seismic Risk Assessment changed to Assessment of physical risk

4.1 Social vulnerability index (SoVI) changed to Results of social vulnerability analysis

4.2 Seismic Risk Assessment changed to Results of seismic risk assessment

4.3 Integrated Risk Assessment to Results of Integrated risk assessment

**A. Reply to the general comments:**

→ We agree that there are grammatical errors within the text, we have tried to correct them to the fullest. The words "methods" and "method" that were mentioned in the paper have been substituted. We have added a big section called "Discussion" prior to conclusion. The detailed comparison of the study with past studies has been carried out. In addition, we have also included limitations and steps to overcome these setbacks.

**B.** Reply to the specific comments:**

**1. Abstract**

 $\rightarrow$ The abstract has been reformulated. All of the points are addressed in the reformulated version of abstract.

**2. Introduction**

→The introduction has been organized as per suggestions mentioned above. Comment 2.1: The national and global aspects of earthquakes have been separated. Comment 2.2: The text "lack of seismic mapping" has been replaced with "lack of social data for analysis and mapping." Comment 2.3 The sentence has been paraphrased. Comment 2.4 The OpenQuake reference has been cited. Comment 2.5 The administrative division of Nepal has been introduced. Comment 2.6 As mentioned, we have rewritten the section according to the comments.

**3. Theory and Background**

→Comment 3.1: Reformulated sentence as per suggestion: "There are many social vulnerability parameters that determine the impact of natural hazards such as socioeconomic status, geographical features, ethnicity (minority), renter, gender, and age (Burton and Silva, 2016)." Comment 3.2: We have removed the subjective statement "There have always been stories of…" Comment 3.3 and 3.4: Reformulated as per suggestion. Comment 3.5: Included in discussion part.

**4. Materials and methods**

→Comment 4.1: Reformulated the first sentence to: "This study assesses seismic risk by combining it with the human dimensions within the hazard zone similar to that in Burton and Silva (2016). This approach is an integrated seismic risk assessment." Comment 4.2.2 & 4.2.3: Table 2 provides the list of all the variables used for social vulnerability assessment. Out of 45 variables, district-wise indicators were represented by 22 variables and each sub-section (municipality and VDCs) was assumed to have uniform value. Among these 45 variables, seven of them were weighted combination of multiple variables as shown in Table 2. These weighted variables were obtained from 54 variables mentioned in Table 3. Therefore, altogether 92 variables (45-7+54) were considered for SoVI index. Comment 4.2.5: Figure 1 has been moved to the introduction as a part of background information and study area description and the background information in this section has been moved to the theory part.

**Comment 4.2.8**: Yes, we have rechecked and we obtained the test value of 0.000, as shown below. Similarly, we found some other studies with same cases. For example, the study by Yogamalar and Samuel  $(2018)^1$ .

| Kaiser-Meyer-Olkin Measure of Sampling Adequacy. |                    | .888       |
|--------------------------------------------------|--------------------|------------|
| Bartlett's Test of
Sphericity                 | Approx. Chi-Square | 258644.625 |
|                                                  | df                 | 1035       |
|                                                  | Sig.               | .000       |

KMO and Bartlett's Test

**Comment 4.2.11:** Reformulated statement: "As presented in the paper Tate (2012), SoVI scores were used in the form of standard deviations (z-scores) or quintiles to emphasize their relative value." All other comments of section 4.2 and 4.3 have been addressed in accordance to the suggestions. Some important changes are: **Comment 4.3.12:** Reformulated statement: "In this study, the building description and data from Census 2011 were used to develop the exposure model without considering the industrial or commercial buildings. In other words, only residential buildings were considered for the exposure model. The exposure models used in the study are part of seismic risk assessment with uncertainties, although present studies like Kalakonas et al. (2020) and Gomez Zapata et al. (2022) have pointed how the epistemic uncertainties embedded in exposure models are propagated throughout the computation of seismic risk. We have considered five types of residential buildings —....." It is also mentioned in "Discussion" section that "The authors are aware of the fact that numerous estimations such as casualties, non-structural damage, business interruption loss, and loss to critical infrastructure may improve the indicator of physical risk. However, only economic losses to buildings were

<sup>1 Yogamalar, I., & Samuel, A. A. (2018). Classification of shared values for educational research. *International Journal of Human Resource Studies*, *8*(2), 47. DOI: 10.5296/ijhrs.v8i2.12818

Figure 5: Fragility curves for a) adobe b) cement bonded c) mud mortar d) wooden buildings

Figure 6: Fragility curves for RCC buildings

---

## Referee Report (RR1)

Dear Sanish Bhochhibhoya and Roisha Maharjan,

Dear Editor Prof. Dr. Heidi Kreibich,

Please find below the comments after having read your revised manuscript.

In general, the authors significantly improved the quality of the first submitted paper. The authors accepted several suggestions, but not as many as I was expecting. I have found that the style of writing the paper is still not mature enough. There are some basic concepts that are mixed up and the paper still follows a quite disorganized structure. Therefore, in my opinion the paper should not be accepted as it is. The paper still requires several modifications and hence it is still in the "major revisions" category. I feel like I have been suggesting changes about the most basic issues, such as terminology, structure, colours, legend, captions, but also about more profound topics such as clear comparisons between the author's results and exiting studies and a compressive discussion (which are still missing). Hence, I honestly feel that the manuscript should have been sent to some colleagues asking for feedback before submitting to the Journal in a first place. I will elaborate more about the former ideas in the following while referring to the formerly listed comments of the first revision:

**A. General comments.**
1. English quality has improved, but there are still some remaining sentences to fix. Hopefully, that can be handled at a later stage (after a new revision, if the editor finds this pertinent).
2. Ok. The suggestion was accepted by the authors. Even though the authors have correctly rewritten the parts where the expression "a method is proposed" (or similar) had been initially stated, there are still several parts where the authors should have better emphasized that many of the inputs used in their study do not come from their own data, models or assumptions, but do come from existing ones (i.e. all that is related to physical seismic risk).
3. Ok. The suggestion was accepted by the authors. A discussion section is presented. However, the content therein is not entirely satisfactory, as will be elaborated later on.
Line 409: "This variability in the result is due to differences in variables and hazards considered during the analysis". The authors used the word "hazards". Why? Was not the seismic ground shaking the only hazard considered? Or do you refer to the distinctive input seismic hazard levels considered by the two mentioned studies? This is distracting.
The following comment is related to the suggestion **B.3.**6: The reference "Schiappapietra and Douglas, 2020" is not a correct citation for the role of spatially correlated ground motions on seismic risk assessment. That study only focused on the physical phenomenon, but not on their effects on risk. Then, this should be removed or relocated. Also, they say "spatially-correlated distribution". *Distribution of what?* The entire sentence is not clear enough and it is presented more as background information (something people would write in an Introduction (*or in the suggested chapter*) and not in a Discussion). Then, the authors say: *"However, in our study, we have used the conventional method of probabilistic seismic risk assessment due to its simplicity"* as if the incorporation of spatially correlated ground motions was an alternative method to PSHA, when in reality, they can be complementary. This shows the lack of understanding of these basic concepts. The OpenQuake engine (used by the authors) has already the model of Jayaram and Baker (2009) included, as well as some of the GEM recent manuals have a short explanation of their importance.
> *Jayaram, N.; Baker, J.W. Correlation model for spatially distributed ground-motion intensities. Earthq. Eng. Struct. Dyn. 2009, 38, 1687–1708.*

The intention of having suggested commenting on this topic in the Discussion section was more focused towards rather acknowledging the related limitations and outlook.

In the discussion section, you have included the sentence: *"The rescaling is necessary to integrate social vulnerability with physical risk although the rescaling of the estimates may have resulted the loss of spatial information of physical damage results"*. I think I do understand what you try to say, but due to the terminology used in the text, "loss" might not be a good word selection. I suggest changing and making the sentence clearer.

4. Ok. The suggestion was accepted by the authors. There is a generalized improvement in the manner this is presented in comparison with the first version.

**B. Specific comments.**

1. "Abstract"
   In general, the quality of the new abstract has been significantly improved with respect to the initial version of the paper. However, its last part is presented more a too detailed summary. Providing a short summary in the abstract is always nice, but not to the degree of mentioning the number of variables. This could be moved to the last part of the introduction.

   There is a persistent imprecision in the abstract. The sentence *"the assets used were five types of buildings under the exposure model"* is not accurate. The exposed assets are not "types", but real objects (in this case residential buildings) that are classified into simplified typologies for the hazard-dependent vulnerability of interest (in this case, seismic ground shaking). In reality, the assets are the Residential buildings of Nepal (classified into five types), not the types.

   Also, the entire sentence *"In this paper, the physical or seismic risk was evaluated from the exposure model, hazard curves, and the vulnerability model of the country"* can be misleading. There are no unique exposure or vulnerability models for any region in the world. Hence, using "the" in "the exposure model" and "the physical vulnerability functions" is incorrect. They are not unique invariable models. Also, since the ones used by the authors basically follow the same building classes and corresponding fragility functions, this should be rephrased to "an existing exposure model for residential buildings". Otherwise, it might lead to the wrong belief that the exposure and vulnerability parts of the paper are your contribution (which is not the case).

   Moreover, the words "hazard curve" do not really fit here. No hazard curve was really presented in the entire paper. The authors do not present any result related to these computations, (only presented existing source models and recalled the selection of GMPE by others. This might lead to the wrong belief that hazard curves will be presented, or even, that a *new* probabilistic seismic hazard assessment will be presented (which is not the case). Therefore, the authors should also state something similar to: "an existing probabilistic seismic hazard assessment".

   **1.1.** Ok. The suggestion was accepted by the authors.
   **1.2.** Ok. The suggestion was accepted by the authors.
   **1.3.** Ok. The suggestion was accepted by the authors.
   **1.4.** This is not sufficient. This is correctly done at the beginning of section 3.2.3, but not in the abstract as requested.
   **1.5.** The inaccurate sentence written in the former version was corrected. However, aligned with the previous comment, the expression "residential buildings" was not accepted in this section by the authors. Also, the explicit request on writing "seismic ground shaking" (as it is the only hazard evaluated) was not accepted either. The latter is very important as explained in the first review.

**1.6.** Ok. The suggestion was accepted by the authors.

2. "1. Introduction"

   New comment: the expression "hazard management" is incorrect. Do you refer to other concepts different from the hazard (e.g. disaster risk)?

   **2.1.** The suggestion was not accepted. I still find that the structure of providing global characteristics in between two subsections with issues about Nepal is disorganized.

   **2.2.** Ok. The suggestion was accepted by the authors.

   **2.3.** Ok. The suggestion was accepted by the authors.

   **2.4.** Ok. The suggestion was accepted by the authors.

   **2.5.** Ok. The suggestion was accepted by the authors.

   **2.6.** Ok. The suggestion was accepted by the authors.

3. "2. Theory and background".

   **3.1.** Ok. The suggestion was accepted by the authors.

   **3.2.** Ok. The suggestion was accepted by the authors.

   **3.3.** Ok. The suggestion was accepted by the authors.

   **3.4.** Ok. The suggestion was accepted by the authors.

   **3.5.** Ok. The suggestion was partially accepted by the authors. Corrections of terminology were done. However, the text of interest was moved to another section. It should have remained here, as the hazard component is part of the "material" used in your work. (See comment B3.6). Nonetheless, it is Ok to leave the issue related to the Poissonian assumption of PSHA in the Discussion.

   **3.6.** The suggestion was partially accepted. No comment on the spatial correlation was provided in one of the advised sections. Something similar was mentioned in the discussion part as discussed above.

   Moreover, sections 3.2.2 and 3.2.1 integrally present the work of others (the seismic zonation, or the selection of GMPE). There is nothing that the authors have done by themselves in those subsections. Also, even though the actual outcome of these two sub-processes is the probabilistic seismic hazard (at certain return periods), there is nothing written about these outcomes in this section. Outcomes such as the acceleration values obtained from hazard curves, either computed by you (during the recalculation of the work of Chaulagain et al, 2015) or by other authors are missing here. It is true that the authors provided those values in the Discussion section (between lines 417 and 422). However, providing such background information at that very last stage (the first time in the whole text that acceleration values manner are mentioned in this new version of the manuscript), is extremely weird and disorganized. Moreover, the probability of exceedance used in your calculations is not clearly stated. I suspect it was 10%, but this guess can be ambiguous by other readers considering that the work you are based on (Chaulagain et al, 2015) also provided results for 2%. Considering these reasons, I recommend that the authors merge Section 3.2.2 and 3.2.1 into a single one: "*Seismic hazard assessment*" or something similar, also including the lines from 417 to 422 (of course, re-writing if needed).

4. "3. Materials and methods".

   **4.1.** Ok. The suggestion was accepted by the authors

   **4.2.** " 3.1. Social vulnerability assessment"

       **4.2.1.** Ok. The suggestion was accepted by the authors

       **4.2.2.** Ok. The suggestion was accepted by the authors. Understood.

       **4.2.3.** Ok. The suggestion was accepted by the authors.

       **4.2.4.** Ok. The suggestion was accepted by the authors.

**4.2.5.** Figure was not relocated as advised.

**4.2.6.** Ok. The suggestion was accepted by the authors

**4.2.7.** Ok. The suggestion was accepted by the authors

**4.2.8.** The suggestion was not accepted by the authors. Although the authors provided an explanation about used test, there is no real reason of using the same outcome of the software (using 3 decimals in 0.000) right after they wrote other numbers with different number of digits.

**4.2.9.** The full name of SPSS was written in the text as suggested. However, it is still missing to provide it within the Reference list. Thus, this suggestion remains incomplete.

**4.2.10.** The suggestion was not accepted by the authors.

**4.2.11.** Ok. The suggestion was accepted by the authors

**4.3.** "3.2. Seismic Risk Assessment"

**4.3.1.** Ok. The suggestion was accepted by the authors

**4.3.2.** Ok. The suggestion was accepted by the authors

**4.3.3.** Ok. The suggestion was accepted by the authors.

**4.3.4.** Ok. The suggestion was accepted by the authors, but the word "similar" does not mean "identical" (your case). I still suggest rewriting this. It is great you have written "Main Himalayan Thrust" in Sect. 2.2.

**4.3.5.** Ok. The suggestion was accepted by the authors.

**4.3.6.** Ok. The suggestion was accepted by the authors

**4.3.7.** Ok. The suggestion was accepted by the authors

**4.3.8.** Ok. The suggestion was accepted by the authors (the figure with the seismic sources was improved). However, commenting/ the newly added information is missing in the text (it still has the same description as before). I had suggested including a sentence that was not accepted to be included.

**4.3.9.** The suggested reference "Rao et al., (2020)" was not cited to support the explicitly suggested topic. Instead, it was cited in a very generic sentence on page #1. These authors were not the first ones to work on such a topic (the one you cite them for on page 1), and their work is neither mostly recognized for that topic. Adding suggested references randomly just add more noise to the paper. This comment also applies for the suggested reference "Gomez-Zapata et al., (2021)" that was used to reinforce a statement about fragility functions (page 4). These two references should be relocated to reinforce the specific topics for which they were suggested in the first review round (their main topic), not generic aspects of any seismic risk assessment. For instance, the central topic of the first one is not "disaster", and the latter one did not propose the concept of "fragility functions", right? You could also consider moving them to the discussion if you feel that the topics exposed could be taken into account in the future (for instance, as an outlook.

**4.3.10.** Ok. The suggestion was accepted by the authors.

**4.3.11.** Suggestion was not accepted by the authors.

**4.3.12.** Ok. The suggestion was accepted by the authors

**4.3.13.** Ok. The suggestion was accepted by the authors

**4.3.14.** Ok. The suggestion was accepted by the authors

**4.3.15.** Ok. The suggestion was accepted by the authors

**4.3.16.** Ok. The suggestion was accepted by the authors. However, there is no need of separating Fig 5 and 6. Fig. 6 could be just listed as Fig."e)". Moreover, in line 287, the authors have written: "After defining fragility functions, it is also important to assess the correlation between the logarithmic means and standard deviations" is not accurate. The fragility functions are implicitly defined by their

logarithmic means and standard deviations. It is not a second step and has nothing to do with "correlation". This must be corrected using a simpler expression.

- **4.3.17.** Ok. The suggestion was accepted by the authors. This is a significant improvement.

- **4.4.** "3.3 Integrated risk assessment"
    - **4.4.1.** Ok. The suggestion was accepted by the authors
    - **4.4.2.** Suggestion was not entirely accepted by the authors. Despite giving explicit hints, the caption was not modified.
        - **4.4.2.1.** The authors chose to do it in the text, not in the caption.
        - **4.4.2.2.** Suggestion was not accepted by the authors.
        - **4.4.2.3.** The authors combined this comment with 4.4.2.4, which is Ok. The suggestion was accepted by the authors. The authors made an attempt to include a similar sentence to the suggested one. However, it is not well written (the words "present" and "although" do not fit there, and the overall sentence sounds incomplete). Hence, due to the way it was presented, it does not yet fulfill the aim that was requested in the first revision.
        - **4.4.2.4.** See above.
        - **4.4.2.5.** Ok. The suggestion was accepted by the authors.
    - **4.4.3.** Ok. The suggestion was accepted by the authors
    - **4.5.** Ok. The suggestion was accepted by the authors
    - **4.6.** Ok. The suggestion was accepted by the authors. For the related paragraph, I still recommend to cross-referencing the sections using parenthesis or chapters in which you presented each component.

5. "Results and discussion"
    - **5.1.** Ok. The suggestion was accepted by the authors and there are two different sections now.
    - **5.2.** "4.2. Seismic Risk Assessment".
        - **5.2.1.** This comment is one of the most relevant suggestions. However, the suggestion was not accepted by the authors. The authors fully rely on the outcomes of the mentioned study in the revision (Chaulagain et al, 2015) not only for the seismic hazard (the only aspect they comment on), but also on the exposure model, fragility functions, and even on the replacement costs and loss ratios (basically everything). In the discussion section, the authors should have commented on this issue as well as what could be results if these components were updated, or if other models were used. As suggested, the authors should be made it clear that these steps are not their contribution, but only the incorporation of the social vulnerability part. Nonetheless, I can clearly see that the results between Chaulagain et al, 2015 (fig 7) and the author's work (fig 13) are different, but once again, the authors do not do such a comparison while discussing why their result is improving that existing study. This suggestion was just neglected.
        - **5.2.2.** Ok. The suggestion was accepted by the authors.
        - **5.2.3.** The suggestion was not accepted by the authors. The figure has the same caption as before.

6. Conclusions
    - **6.1.** Ok. The suggestion was accepted by the authors.
    - **6.2.** Ok. The suggestion was accepted by the authors.

**C. Technical corrections**

1. There is an inconsistency: The authors have included the DOI and year for most of the references. However, the journal names have been deleted from each reference. This is unacceptable.
2. Ok. The suggestion was accepted by the authors
3. Ok. The suggestion was accepted by the authors
4. Ok. The suggestion was accepted by the authors
5. Ok. The suggestion was accepted by the authors
6. Ok. The suggestion was accepted by the authors
7. Suggestions were not accepted by the authors.
8. Ok. The suggestion was accepted by the authors
9. Ok. The suggestion was accepted by the authors

**D. New general comment (second revision)**

The results obtained in this study should be freely available to the reader. Please make sure to provide one or several data repositories with the input data and outputs. This should be cited in the manuscript. This is because the results should be transparent and reproducible.

**E. New Specific comments (second revision).**

I think that figure 4b is not informative enough for the purposes and aims of exposure modelling at this regional scale. Although box whisker plots might be informative to visualise the median and extreme values, this might not be that interesting for exposure modelling. It is evident the authors wanted to do an alternative plot to Fig 3b provided by Chaulagain et al, 2015. However, the percentages shown by the referred paper are more interesting. In this sense, please note that the values you report in line 282: "135.73, 603.36, 239.91, 340, 46.33" as "average" of the types RCC with pillar, Mud-Bonded, Cement-Bonded, Wooden-pillar, and Adobe respectively cannot easily "read" from that figure, specifically for RCC with pillar and Abode types. This is more a "form" comment. Please be aware that the total number of buildings per class is more interesting than the median values across the administrative divisions. You can think of including the respective percentages and total building counts per class.

**F. New technical corrections (second revision)**

1. The authors use sometimes "MIN-MAX", "MINMAX", "Min-Max". Please select only one notation and harmonize.
2. Line 330: "that affect the earthquake risk". The verb "affect" is out of context here. This sentence is in general not needed and actually can lead to confusion: as if the 2 mentioned variables might induce to the modification of something pre-established, which is not the case.
3. The caption of Figure 13 is not comprehensive. It must be more descriptive and self-explanatory.
4. The quality of most of the figures that remained the same (e.g. Fig 10, 11 in the new version) has decreased in comparison with the ones in the initially submitted version (i.e. Fig. 7 and 8). Please improve the colour quality of those figures.
5. Line: 442: "doesn't" is not proper English. Please change it.

---

## Author Response (AR2)

We are very grateful for the extensive review of our paper including comments and suggestions. A revision of the paper has been carried out to take account of all the comments.

**Reply to Referee:**

1. Seismic hazard, exposure, and physical risk:

   The seismic risk assessment of the study is indeed based on the works by Chaulagain et. al (2015) to some extent. We adopted the source zone (Chaulagain et. al (2015) also adopted the source zones from the study by Thapa and Guoxin (2013)) and fragility models from their study. However, there are two major changes in our research. The building distribution (exposure model) of our works is at VDC/Municipality level in comparison to the district level study in Chaulagain et al. (2015). Also, we calculated per-capita economic loss for each VDC/Municipality as our result, while Chaulagain et. al (2015) computed the total economic loss for each district in their study.

   Understanding the ethical values of the research, we, the authors, have acknowledged other researchers' work wherever necessary, and had no intention to mention others' work as ours. After the second revision, we have increased the number of mentions of Chaulagain et. al works in the reviewed manuscript to make sure the input models, adopted from their study, are not misinterpreted as our idea. The list of the acknowledges of Chaulagain's works are as follows:

   - Line 69-70: "In this study, the country-level earthquake risk estimates from the Global Earthquake Model OpenQuake (Pagani et al., 2014) were analysed by using the input models (seismic hazard sources, fragility functions, and consequence model) given by Chaulagain et al. (2015)."
   - Line 399-400: "On the other hand, the seismic source model, fragility curves, and consequence model used in the study by Chaulagain et al. (2015) were used to evaluate the physical risk in OpenQuake. Similar to the study by Burton and Silva (2016), the integrated risk was evaluated using integrated risk modelling toolkit."
   - Line 265-266: "We assumed the tectonic region as a shallow crust and subduction interface like that in Chaulagain et al (2016)."
   - Line 255-256: "In this study, the twenty-three source zones similar to that of Thapa and Guoxin (2013) were considered for probabilistic seismic hazard analysis."
   - Line 284-285: "In this study, the fragility model developed by Chaulagain et al. (2015) was adopted for different building types."

2. Discussion remains not well organized.

   We had completely added this section after the first major revision, and have revised and structured this section, as per the suggestion by reviewer. The major changes are:
   - ➔ We have subdivided the discussion into three sections:
     1. Discussion on social vulnerability assessment
     2. Discussion on physical risk assessment
     3. Discussion on integrated risk assessment

3. Reply to the Reviewer 2 on first major revision comments

   The authors are grateful for the extensive review. We had addressed all the points with our best effort, and are described in detail in our first reply (Section A, B, C: Reply to the general, specific, and technical comments).